# Direct measurements of the colloidal Debye force

Hyang Mi Lee[1,6], Yong Woo Kim[1,6], Eun Min Go[2,4,6], Chetan Revadekar[1], Kyu Hwan Choi[1,5], Yumi Cho[2], Sang Kyu Kwak [3] ✉ & Bum Jun Park [1] ✉

Colloids often behave in a manner similar to their counterparts in molecular space and are used as model systems to understand molecular behavior. Here, we study like-charged colloidal attractions between a permanent dipole on an interfacial particle and its induced dipole on a water-immersed particle caused by diffuse layer polarization. We find that the scaling behavior of the measured dipole-induced dipole (D–I) interaction via optical laser tweezers is in good agreement with that predicted from the molecular Debye interaction. The dipole character propagates to form aggregate chains. Using coarse-grained molecular dynamic simulations, we identify the separate roles of the D–I attraction and the van der Waals attraction on aggregate formation. The D–I attraction should be universal in a broad range of soft matter, such as colloids, polymers, clays, and biological materials, motivating researchers to further conduct in-depth research on these materials.

The study of electric dipoles among molecules provides important information about molecular assemblies and the microstructure formation of proteins and RNA/DNA[1–4]. For polar molecules, a permanent dipole moment is generated by variations in electron density (i.e., the difference in electronegativity). The permanent dipole $\mu_1$ can induce a dipole moment $\mu_2^*$ in a polarizable nonpolar molecule (Fig. 1a); the induced dipole moment temporarily arises from a dipole-generated electric field that distorts the electron distribution and nuclear position in the molecules[5]. The dipole-induced dipole (D–I) interaction, which is referred to as the Debye interaction, is always attractive. It is given by $U_{D-I} = U_{Debye} = -\frac{\mu_1^2 \alpha_2'}{4\pi\varepsilon_0\varepsilon_r^2} r^{-6}$, where $\varepsilon_0$ is the vacuum permittivity, $\varepsilon_r$ is the dielectric constant of a medium, $\alpha_2'$ is the polarizability volume, and $r$ is the separation between the two interacting molecules[6,7]. Typically, the polarizability volume of a molecule $\alpha_2'$ is approximately equal to its molecular volume[7].

Colloidal particles often behave in a manner similar to their counterparts at the molecular scale[8–11]. Accordingly, they have been used to study complex behaviors of atoms and molecules because the

time and length scales can be adjusted to levels that can be measured directly with conventional instruments[12]. In particular, the fundamental interaction phenomena of molecules are similar to those of colloids, while the magnitude of colloidal interactions is significantly larger, allowing direct measurements of the interaction force. For example, an induced dipole could be formed when the diffuse layer around a dielectric particle in a medium is polarized along the direction of an applied external field[13–17]. When the external field is continuously applied, the induced dipole caused by diffuse layer polarization is maintained. Therefore, the induced dipole behaves as a permanent dipole with a fixed orientation, and the measured interparticle interaction decays as $r^{-3}$ [18]. When colloidal particles reside at a fluid–fluid interface (e.g., oil–water or air–water)[19], asymmetric surface charge dissociation occurs across the interface, leading to the formation of an electric dipole perpendicular to the interface[20–22]. The lateral electrostatic repulsion between the interface-trapped particles decays as $r^{-3}$, which also accounts for the orientation-fixed dipole-dipole interaction model. The $r^{-3}$ dependence due to dipole-dipole colloidal

[1]Department of Chemical Engineering (BK21 FOUR Integrated Engineering Program), Kyung Hee University, Yongin, Gyeonggi-do 17104, South Korea. [2]School of Energy and Chemical Engineering, Ulsan National Institute of Science and Technology, Ulsan 44919, South Korea. [3]Department of Chemical and Biological Engineering, Korea University, Seoul 02841, South Korea. [4]Present address: Corning Technology Center Korea, Corning Precision Materials Co., Ltd., 212 Tangjeong-ro, Asan, Chungcheongnam-do 31454, South Korea. [5]Present address: Department of Chemical Engineering, University of California Santa Barbara, Santa Barbara, CA 93106, USA. [6]These authors contributed equally: Hyang Mi Lee, Yong Woo Kim, Eun Min Go. ✉e-mail: skkwak@korea.ac.kr; bjpark@khu.ac.kr

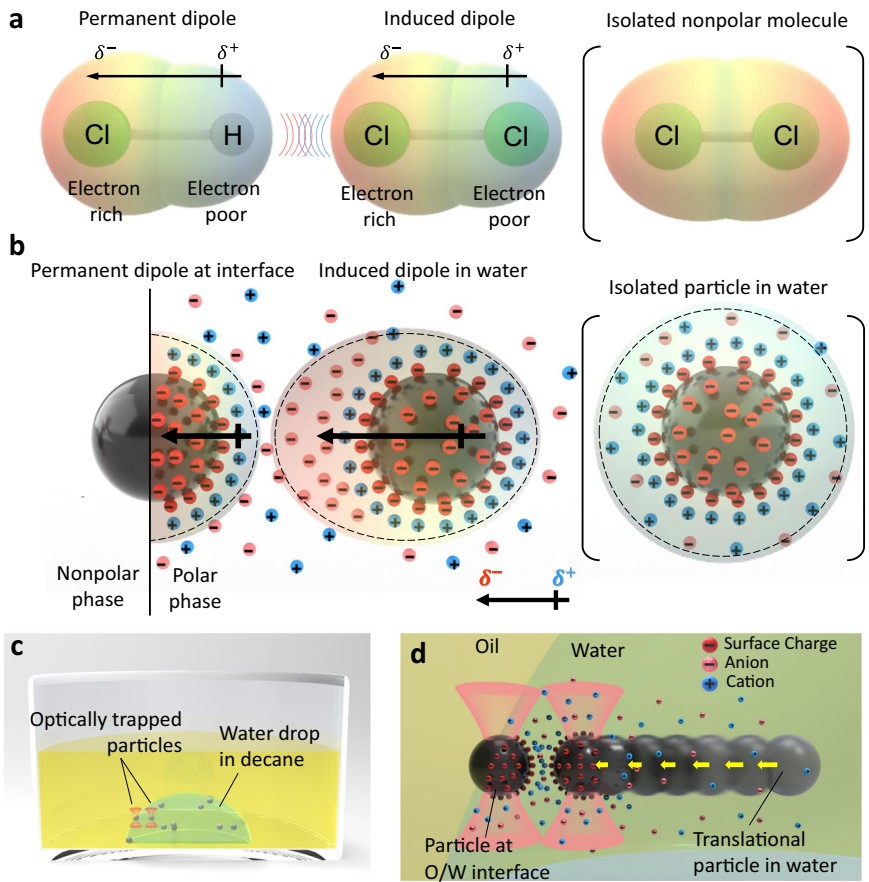

**Fig. 1 | Schematic illustration. a** Molecular dipole–induced dipole (D–I) interactions between polar and nonpolar molecules. **b** Colloidal D–I interactions between an interface-trapped particle and a water-immersed particle. **c** Experimental fluid cell (not drawn to scale). **d** Direct measurements of the colloidal D–I interaction via optical laser tweezers.

interactions has been reported, whereas experimental studies regarding the D–I interactions (i.e., the colloidal Debye interaction with the $r^{-6}$ dependence) have not.

Here, we present the direct measurements of colloidal D–I interactions using optical laser tweezers. We postulated that the dipole-generated electric field around colloidal particle trapped at an oil–water interface can lead to ion rearrangements or diffuse layer polarization around another approaching particle in water (Fig. 1b). The diffuse layer polarization causes an induced dipole around the water-immersed particle, and the corresponding D–I interaction between the two like-charged particles should be attractive. In this regard, the classical electrostatic double-layer interaction, which is typically repulsive for like-charged particles, should be altered or diminished. When the two particles near each other, the ion rearrangement between them is spatially limited. Consequently, the D–I attraction between them likely decreases. If these events occur simultaneously or subsequentially and the two particles are sufficiently close to the vdW-dominant region, they will spontaneously form an aggregate dimer. Furthermore, if the dipole character of the dimer is maintained, it will be possible to form a multiparticle-composed colloidal string by successively approaching multiple particles in water into the proximity of the interface-trapped particle using optical laser tweezers.

## Results

### Direct measurements of the D–I interaction force

To measure the interaction force between an interface-trapped particle and a water-immersed particle, we prepared a sessile water drop with polystyrene microspheres having surface sulfate groups (SPS particles)

in an n-decane environment (Fig. 1c). The particles dispersed in water were negatively charged with a $\zeta$-potential of $\psi = -57.5 \pm 2.2$ mV, and the average particle diameter was $d = 2.96 \pm 0.05$ μm (Supplementary Table 1). The three-phase contact angle of the SPS particle at the oil–water interface was ~99.4°[23]. Using time-sharing optical laser tweezers (Supplementary Note 1) and the drag calibration method (Supplementary Fig. 1; Supplementary Notes 2 and 3)[23–25], one particle ($P_1^i$) was attached to the oil–water interface, and another particle ($P_2$) in water was translated stepwise toward $P_1^i$ to measure the interaction force (Figs. 1d, 2a–c). The superscript i of $P_1^i$ denotes interface attachment. We used 10 mM NaCl water to successfully attach $P_1$ to the interface; the electrolyte reduced the electrostatic disjoining pressure between a colloidal particle and a fluid–fluid interface (both negatively charged), facilitating its interface adsorption (Supplementary Movies 1 and 2)[26,27]. The presence of electrolyte also could provide a means of inducing ion rearrangement and formation of an induced dipole around $P_2$ in water. Based on our preliminary experiments, we confirmed that the particles dispersed in 10 mM NaCl water formed few particle aggregates, even if we tried to force them together using optical tweezers. We will discuss this later.

Quantitative measurement of the interaction force between $P_1^i$ and $P_2$, which are like-charged, showed attraction. We measured the forces for 290 pairs due to the interaction heterogeneity present when measuring the colloidal interaction forces[22,23,28]. Among these pairs, the 72 force profiles ($F_{D–I}$) with considerable attraction magnitude are shown in Fig. 2d. The other pairs did not show measurable attractive forces. The measured force was fitted with $F_{fit} = F_0\left(\frac{d}{r}\right)^b$. Two-parameter fitting (insets in Fig. 2d) resulted in average values of $\langle F_0 \rangle = -0.44 \pm 0.17$ pN and $\langle b \rangle = 7.18 \pm 1.22$ with $\langle \chi^2 \rangle = 9.87 \times 10^{-3}$.

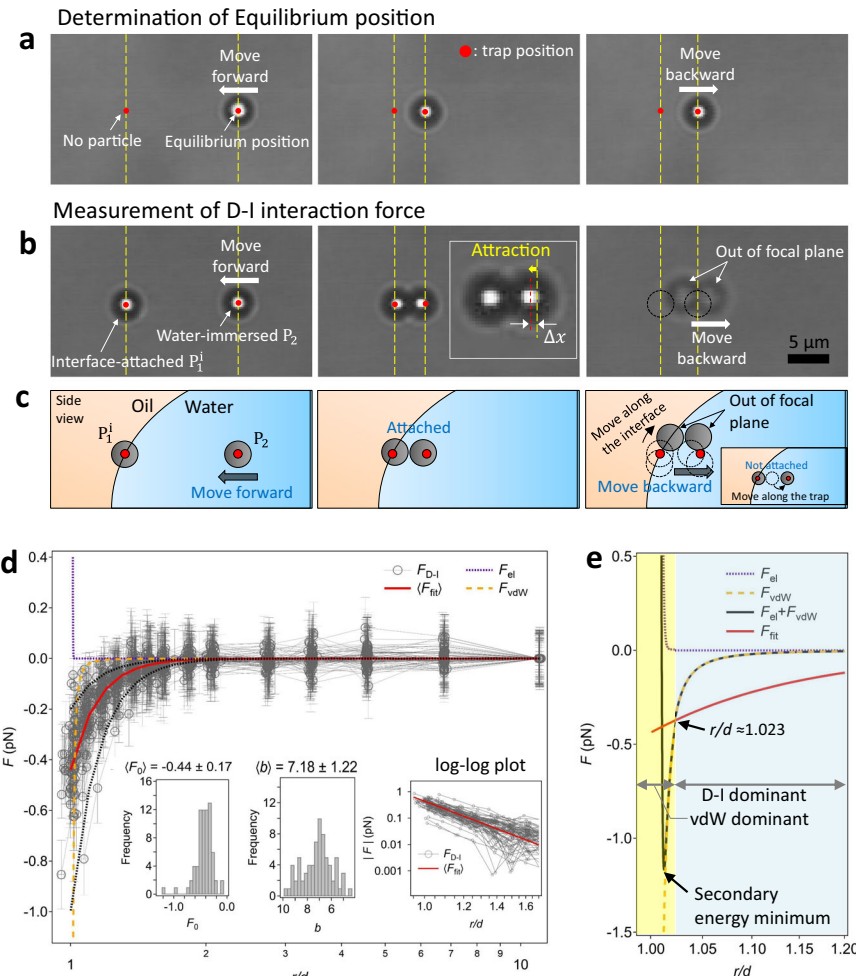

**Fig. 2 | Direct measurements of the D–I interaction force. a** Determination of the equilibrium position of $P_2$ in water. **b** Measurement of $F_{D-I}$. Red dots indicate the trap positions (not drawn to scale). **c** Schematic side view for the force measurement and the confirmation procedure of dimer formation (not drawn to scale). **d** Measured D–I interaction force profile ($F_{D-I}$) and comparison with the DLVO forces ($F_{el}$ and $F_{vdW}$). The water phase contains 10 mM NaCl. The error bar in each force profile indicates thermal fluctuations of a trapped particle. The $x$ axis of the graph is on a logarithmic scale. The red solid line represents a fitted curve that uses

the mean values of the two fitting parameters, $\langle F_0 \rangle = -0.44$ and $\langle b \rangle = 7.18$. The black dotted lines indicate the guideline for $F \sim r^{-7}$. The force profiles display before the paired particles come into contact. The left two inset plots display histograms of the values of the two fitting parameters for 72 pairs. The inset on the right represents a log–log plot of $|F|$ versus $r/d$ for short-range separation. **e** Magnified force profiles in the short-range separation, and the $x$ axis is on a logarithmic scale.

Accordingly, the result demonstrated that the attractive force and the corresponding interaction potential decay as $F_{D-I} = F_{fit} \sim r^{-7}$ (red solid line and dashed guide lines in Fig. 2d) and $U_{D-I} \sim r^{-6}$, respectively, where the power law exponent of –6 in the interaction potential indicates the signature of the D–I interaction in molecular space.

The D–I interaction force can be expressed as $F_{D-I} = -\frac{dU_{D-I}}{dr} = F_0 \left(\frac{d}{r}\right)^7$, where $F_0 = -\frac{3\mu_1^2 \alpha_2'}{2\pi\varepsilon_0 \varepsilon_w d^7}$ and $\varepsilon_w$ represents the water dielectric constant (Supplementary Note 4). The permanent dipole moment $\mu_1$ for the interface-trapped particles can be obtained using the self-potential method[23] by measuring the dipole–dipole pair interaction forces between $i$ and $j$ particles at a planar oil–water interface $F_{D-D} = \frac{3\mu_i \mu_j}{8\pi\varepsilon_0 \varepsilon_{oil} d^4} \left(\frac{d}{r}\right)^4$, where $\varepsilon_{oil}$ is the decane dielectric constant (Supplementary Fig. 2a)[20,29]. The mean value for 18 particles was found to be $\langle \mu_1 \rangle \times 10^4 = 4.5 \pm 2.0$ pC·μm at the same fluid condition (Supplementary Fig. 2b, c; Supplementary Note 5). The polarizability volume was then calculated as $\alpha_2' = -\frac{2\pi\varepsilon_0 \varepsilon_w d^7 F_0}{3\mu_1^2} = \alpha_{2,exp}' \approx 6.29$ μm³ at $F_0 = -0.44$ pN, which was in order-of-magnitude agreement with the theoretical prediction of the molecular polarizability volume in vacuum,

$\alpha_{2,theory}' \sim \frac{4\pi}{3} \left(\frac{d}{2}\right)^3 \approx 13.6$ μm³. Additionally, a dielectric sphere in a solvent medium with dielectric constants $\varepsilon_p$ and $\varepsilon_w$ can be polarized by an electric field, and the effective polarizability volume $\alpha_{eff}'$ can be reduced by a factor of $\left|\frac{\varepsilon_p - \varepsilon_w}{\varepsilon_p + 2\varepsilon_w}\right|$[7], resulting in $\alpha_{eff}' \approx 6.47$ μm³, which shows excellent agreement with the experimental value $\alpha_{2,exp}'$.

When two particles were immersed in water with $C_{NaCl} = 10$ mM, they did not show such significant attraction (Supplementary Fig. 3). In contrast, the classical Derjaguin-Landau-Verwey-Overbeek (DLVO) interaction represented the presence of a secondary energy minimum at $\frac{r}{d} = 1.012$ with a well depth of $F_{well} = -1.17$ pN (Fig. 2e; Supplementary Note 6). This discrepancy between the experimental results and theoretical prediction is likely due to the surface roughness effect for typical colloidal particles, which can increase both the double layer repulsion ($F_{el}$) and the vdW attraction ($F_{vdW}$, mainly in shorter range separations), resulting in pushing the secondary energy minimum to larger distances and decreasing the well depth[30–32]. The measured scaling behavior of the attractive force was $F_{fit} \sim r^{-7}$ (Fig. 2d) when two particles were present at the interface and in water, verifying that the

double layer repulsion should be replaced with D–I attraction over a certain range of separations.

## Determination of dimer formation probability

We determined the probability of forming an aggregate dimer of $P_1^i$ and $P_2$ while varying a holding separation $r_h/d$ and holding time $t_h$. Two particles were brought into close proximity and held at $r_h/d$ for $t_h$, and we attempted to detach them by translating $P_2$ backward (Supplementary Fig. 4). Due to the interaction heterogeneity[22,23,28], we repeatedly performed the dimer formation experiments at the same conditions. As shown in Fig. 3a, the dimer formation probability $P_f$ was significantly affected by $r_h$. For the conditions of SPS, $C_{NaCl} = 10$ mM, and $t_h = 180$ s, there were no dimers formed at $\frac{r_h}{d} \approx 1.16$, and $P_f$ increased as $r_h$ decreased. This result was consistent with the attractive force (Figs. 2d, 3a), in which a stronger D–I force was generated for closer particle separation. In addition, $t_h$ strongly influenced $P_f$. As shown in Fig. 3b, the cumulative $P_f$ increased with $t_h$ and approached a plateau around $t_h = 180$ s at conditions of SPS, $C_{NaCl} = 10$ mM, and $\frac{r_h}{d} \approx 1.04$. Higher values of $t_h$ increased the probability of attraction, which suggests that a certain amount of time might be required to allow ion rearrangement needed to generate D–I attraction. However, the cumulative $P_f$ value did not reach 1, possibly due to the interaction heterogeneity[22,23,28] resulting from the surface charge nonuniformity of the particles[33,34].

We also evaluated $P_f$ for different types of particles (SPS, CPS (carboxyl-PS), APS (amino-PS)) and electrolytes (NaCl, CaCl₂, LiCl). The D–I interaction similarly occurred with $P_f = 0.8$ (8/10) for CPS and $P_f = 0.5$ (5/10) for APS at $C_{NaCl} = 10$ mM, $\frac{r_h}{d} \approx 1.054$, and $t_h = 180$ s (Fig. 3a). The use of different electrolytes (i.e., 10 mM CaCl₂ and 10 mM LiCl) and lower NaCl concentration (1 mM) also led to dimer formation of SPS at $t_h = 180$ s (Fig. 3a, c), $P_f = 0.27$ (4/15) for $C_{CaCl_2} = 10$ mM ($\frac{r_h}{d} = 1.048$), $P_f = 0.2$ (2/10) for $C_{LiCl} = 10$ mM ($\frac{r_h}{d} = 1.048$), and $P_f = 0.3$ (3/10) for $C_{NaCl} = 1$ mM ($\frac{r_h}{d} = 1.040$). The reduction in $P_f$ with a decrease in $C_{NaCl}$ (Fig. 3c) indicates the important role of the surrounding ions in forming the induced dipole on the water-immersed particle. Conversely, when the SPS particles were completely immersed in water and brought as close as possible using optical laser tweezers, $P_f$ was 0 (0/10) at $C_{NaCl} = 1$ mM and 0.02 (6/256) at $C_{NaCl} = 10$ mM (Fig. 3c), which is consistent with the measured force profile without showing a considerable well depth (Supplementary Fig. 3). $P_f$ increased significantly in the water-only environment with higher NaCl concentrations (Fig. 3c), as $F_{el}$ was sufficiently screened, leading to a relatively greater contribution of the vdW force to the dimer formation. Although it was expected that $P_f$ would also increase in such high NaCl decane/water environments, determining the relative contributions of the D–I interaction and the reduced $F_{el}$ to dimer formation in such conditions would be challenging, and it would not be appropriate to claim the

presence of D–I interaction. Based on these results, it is assumed that various factors such as surface functional groups, surface roughness, and ion types and mobility could affect the D–I interaction and dimer formation probability. Therefore, quantitative in-depth investigations on each factor should be conducted in subsequent studies.

## Coarse-grained molecular dynamic simulations

We experimentally demonstrated the presence of the D–I interaction between the interfacial particle and the water-immersed particle at relatively long-range separations. To further support this, we performed coarse-grained molecular dynamic (CGMD) simulations (Supplementary Note 7; Supplementary Fig. 5) to answer several questions. These questions included whether a dipole forms around $P_1^i$ at the oil–water interface, whether ion rearrangement or diffuse layer polarization occurs around the water-immersed $P_2$ due to the $P_1^i$ dipole, whether this ion rearrangement can attract the two particles to each other, and whether the presence of D–I interaction can be confirmed by comparing the particles' behavior in water-only and decane/water environments. The simulations may also provide insight into whether the D–I interaction persists as the particles get closer or whether it disappears at a certain distance due to sparse ion distribution between them. If the two particles are nearly in contact, the diffuse layer polarization around $P_2$ may be reduced, and they may primarily attach to each other under the influence of vdW.

Using CGMD simulations, we confirmed that an isolated SPS particle in water and at the oil–water interface formed a double layer composed of a condensed ion layer and a diffused ion layer (from the particle surface exposed to water), as in Supplementary Fig. 6. The fitted particle diameter at the oil–water interface was ~17.8 nm. We also determined that the $\zeta$-potential of the simulated PS particle was $\zeta_{MD} = -48.2$ mV, which was slightly lower than the experimental value (i.e., −57.5 mV) (Supplementary Fig. 7). Based on these simulation conditions, when the PS particle was at the oil–water interface, the three-phase contact angle was ~70.8° (Supplementary Fig. 8) and the dipole moment of $P_1^i$ perpendicular to the interface was $\mu_{1,MD} = 7.12 \times 10^{-9}$ pC · μm. Due to the proportional relationship between $\mu$ and particle surface area $S_p$[35,36], we normalized $\mu_{1,MD}$ with the surface area ratio $\gamma = \frac{S_{p,exp}}{S_{p,MD}} = 2.65 \times 10^4$. The normalized dipole strength was $\mu_{1,nor} \approx \gamma \mu_{1,MD} = 1.89 \times 10^{-4}$ pC · μm, which was in good agreement with the experimental mean value (i.e., $\langle \mu_1 \rangle = 4.5 \times 10^{-4}$ pC · μm). The slightly lower $\mu_{1,nor}$ could be attributed to the lower $\zeta_{MD}$ value than the experimental one. Then, we used this simulated PS particle to further investigate the D–I interaction.

The dipole moment on the $P_1^i$ at the oil–water interface led to attraction with a water-immersed particle $P_2$. Here, $P_2$ spontaneously

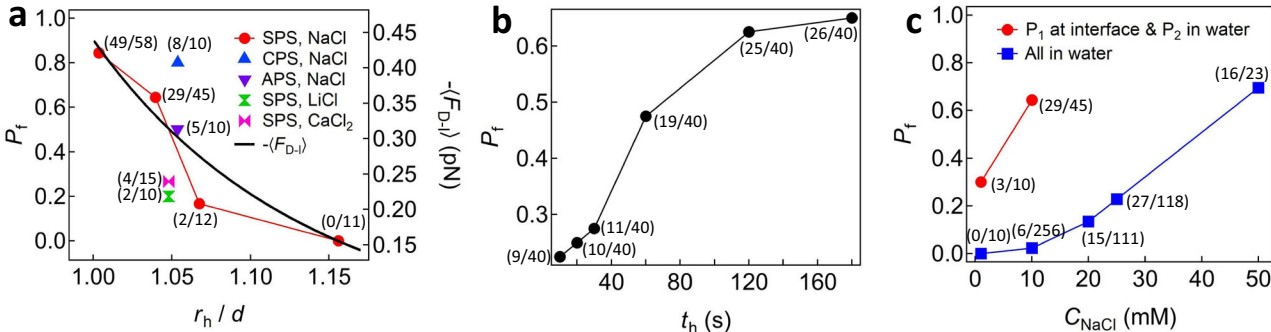

**Fig. 3 | Determination of dimer formation probability $P_f$. a** Dimer formation probability $P_f$ with variations in holding separation $r_h$, types of particles (SPS, CPS, and APS), and electrolytes (NaCl, LiCl, and CaCl₂) at 10 mM concentration. The holding *time* was $t_h = 180$ s. The black solid line indicates that $-\langle F_{D-I} \rangle = -\langle F_{fit} \rangle$ in

Fig. 2d. **b** Cumulative $P_f$ depending on $t_h$. **c** Comparison of $P_f$ when $P_1$ was at the interface and $P_2$ was in water with the case when all particles were in water at different NaCl concentrations.

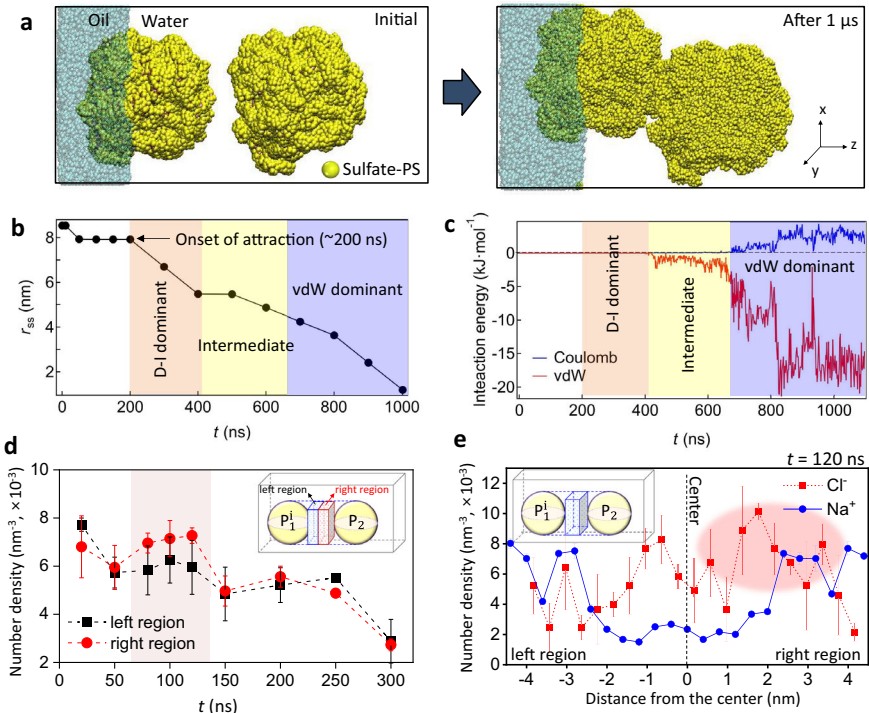

**Fig. 4 | Simulation of $P_1^i - P_2$ dimer formation. a** Initial and final configurations upon dimer formation. Symbols indicating water molecules and ions were omitted for clarity. **b** Surface-to-surface distance $r_{ss}$ between $P_1^i$ and $P_2$ with the simulation time. **c** Interparticle interaction energy. **d** Number density of $Cl^-$ ions in the left and right regions between the two SPS particles. **e** Number density of $Cl^-$ and $Na^+$ between the two particles at $t = 120$ ns. The black dashed line represents the center of the two particles, and the pink area shows a $Cl^-$ rich region. The error bars in panels **d**, **e** indicate three independent simulation runs, and the insets represent the analyzed regions.

approached $P_1^i$ and formed an aggregate dimer (Fig. 4a, b). The interparticle interaction energy was traced during spontaneous attachment (Fig. 4c). The vdW interaction appeared at ~400 ns, whereas the attraction had already occurred at ~200 ns (Fig. 4b). After ~650 ns, the vdW factor became significant and Coulomb repulsion was observed (Fig. 4c). We assumed that the attraction could be initiated by the D–I interaction in relatively longer-range separations (~200–400 ns) where the vdW contribution was negligible, followed by an intermediate state where the D–I and vdW factors co-existed (400–650 ns). Eventually, the vdW interaction became dominant (>650 ns). To demonstrate the presence of the D–I attraction, we captured the ion distribution around the center region between $P_1^i$ and $P_2$ (Fig. 4d, e) and found that the $Cl^-$ distribution was skewed toward $P_2$ for 80–120 ns. Interestingly, the asymmetric charge distribution occurred prior to the onset of attraction at $t$ ~ 200 ns (Fig. 4b), the point at which the $P_2$ began to approach the other particle. Then, the $Cl^-$ ion number density between the two particle regions decreased with time, indicating that the ions diffused out as the particles approached each other. The sparse ion distribution between the particles could have contributed to the reduction in the D–I attraction.

The simulation results demonstrated that the D–I interaction played a critical role in initiating attraction in long-range separations, whereas vdW attraction became dominant in short-range separations, consistent with the experimental observations and the theoretical predictions from the DLVO interactions discussed in Fig. 2d,e. To further investigate the importance of the D–I interaction, a CGMD simulation was conducted for two water-immersed particles using a similar initial condition to that used in Fig. 4a, where the particles were initially separated with a finite surface-to-surface distance. The results showed that the asymmetric anion distribution was not observed (Supplementary Fig. 9), and thus, attraction and dimer formation did not occur in the water-only

condition, highlighting the crucial role of the D–I interaction in initiating attraction over long-range separations at the oil–water environment. In contrast, when the two particles were initially brought as close as possible, the preformed dimer remained stable (Supplementary Fig. 10), indicating that the initial state of the simulation had already fallen into the secondary energy minimum.

Importantly, the dipole character was maintained around $P_2$ of $P_1^i - P_2$ dimer (Fig. 5a). Therefore, we further investigated whether the induced dipole on $P_2$ can generate an induced dipole around another water-immersed particle $P_3$ near the dimer. As shown in Fig. 5b,c, the dimer attracted $P_3$, forming a $P_1^i - P_2 - P_3$ trimer. Similar to the dimer formation in Fig. 4, vdW attraction was weakly observed at ~200 ns, and a significant increase in vdW magnitude was observed at ~350 ns (Fig. 5d). Notably, the spontaneous approach of $P_3$ toward the dimer had already occurred at ~100 ns (Fig. 5c) before the vdW attraction appeared and became dominant. Similar to the dimer formation in Fig. 4, the skewed $Cl^-$ distribution toward $P_3$ was also observed over ~30–100 ns (Fig. 5e) (i.e., prior to the spontaneous approach). This consistently demonstrates the presence of long-range D–I attraction that can be attributed to ion rearrangement. The presence of the induced dipole on $P_2$ of the trimer (Fig. 5a) suggests the possibility of forming longer colloidal aggregate chains.

## Formation of colloidal chains and further demonstration of D–I interactions

Based on the simulation results for forming the trimer, we believed that the dipole character can propagate along the direction of the dipole-induced electric field. Therefore, we attempted to form a trimer by bringing a water-immersed $P_3$ particle toward a $P_1^i - P_2$ dimer using optical laser tweezers. As shown in Fig. 6a, we confirmed the formation of a $P_1^i - P_2 - P_3$ trimer. Furthermore, when the optical trap on either $P_2$ or $P_3$ was removed, it stayed at its position without escaping. A similar experiment was performed using three particles that were

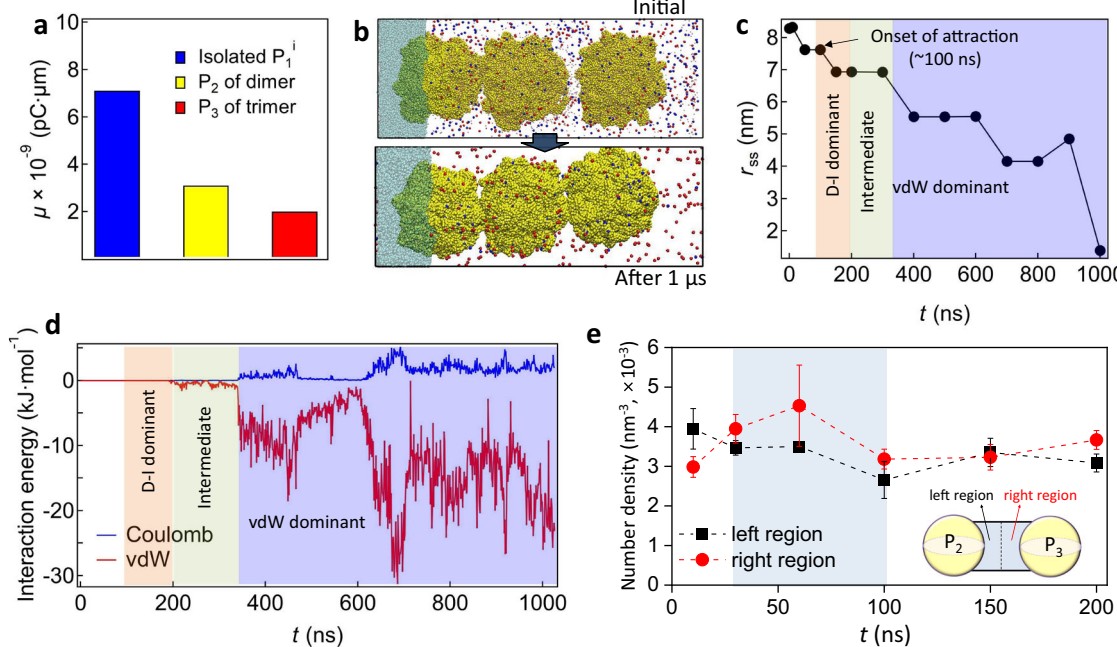

**Fig. 5 | CGMD simulations for the trimer formation. a** Dipole strength of isolated $P_1^i$, $P_2$ of a dimer, and $P_3$ of a trimer. **b** Initial and final configurations when $P_3$ in water approaches the $P_1^i - P_2$ dimer. **c** Surface-to-surface distance between $P_2$ and $P_3$ over the simulation time. **d** Interaction energy between $P_2$ and $P_3$. **e** Number density difference of Cl⁻ ions in the left and right regions between the two particles. The error bar indicates three independent simulation runs, and the inset represents the analyzed region.

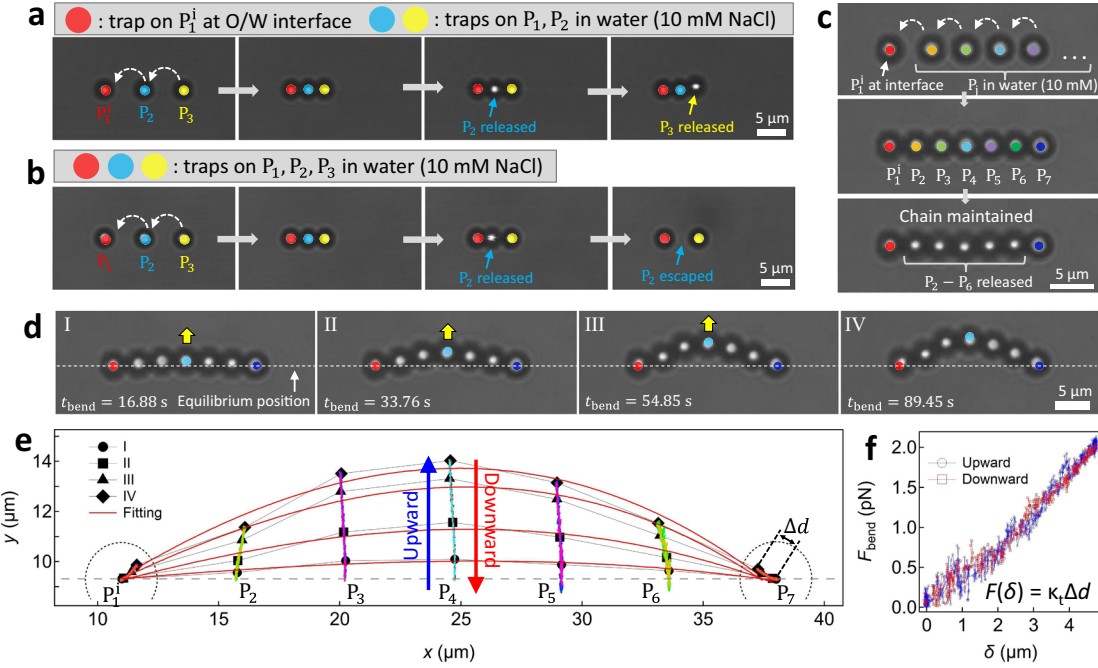

**Fig. 6 | Demonstration of multimer formation. a**, **b** Attempt to form trimers when $P_1^i$ was at the interface (Supplementary Movies 5–7) and all particles were in water (Supplementary Movie 8). **c**, **d** Formation of heptamer and its bending micromechanics (Supplementary Movies 9 and 10). **e** Particle trajectories while $P_1^i$ and $P_7$ of the heptamer were fixed and $P_4$ moved upward and downward successively. **f** Corresponding force $F_{bend}$ as a function of deflection $\delta$. Colors on individual particles indicate the optical traps (not drawn to scale).

completely immersed in water, and removal of an optical trap from any particles led to particle escape (Fig. 6b).

We could build longer colloidal chains by bringing the water-immersed particles one by one to the interface-attached particle. As shown in Fig. 6c, several particles in water were successively attached to the interfacial $P_1^i$ to eventually form a heptamer composed of seven particles. Upon removing the optical traps from the $P_2 - P_6$ and holding those at $P_1^i$ and $P_7$, the linear shape of the chain was maintained. The same experiment was performed but all particles were in 10 mM NaCl water, and no particle attachment occurred

(Supplementary Fig. 11). In addition, the heptamer was subjected to a bending moment by moving $P_4$ upward and holding $P_1^i$ and $P_7$ stationary, in which the optical traps on the other particles were removed. The particle positions from several selected images during bending (I–IV in Fig. 6d) showed good agreement with the shape predicted from the Euler-Bernoulli equation (the red solid curves in Fig. 6e)[37–39]. This result was consistent with a previous report, in which singly bonded colloidal aggregates induced by the vdW force supported significant shear forces[37]. The trajectories of all the particles were similar in both directions when $P_4$ of the heptamer moved upward and downward successively. In addition, the corresponding force could be obtained by using $F_{bend}(\delta) = \kappa_t \Delta d$, where $\delta$ is the chain deflection and $\Delta d$ is the displacement of the $P_7$ particle from its equilibrium position. The resulting profiles during the bending and relaxing events were similar (Fig. 6f), indicating no small-scale rearrangements between the particles due to a critical bending moment. The bending rigidity of the colloidal chain was estimated to be $\kappa_{chain} \approx 0.41$ pN/μm by linear regression of the force profile in Fig. 6f. Using the Johnson–Kendall–Roberts (JKR) theory for particle adhesion, we estimated the Young's modulus of the PS chain to be ~0.05 GPa, which was comparable to those of e.g., polystyrene foam (~0.005 GPa) and low density polyethylene (~0.2 GPa) (more details can be found in Supplementary Note 8)[37,38]. Local rearrangements between the particles and chain rupture were observed when a critical bending moment was applied to a colloidal chain in another bending experiment (pentamer in Supplementary Fig. 12). These results indicated that the particles was in a shallow secondary energy minimum formed by the colloidal vdW force that could be detached by optical tweezers[32]. Overall, the results consistently demonstrated the separate roles of D–I and vdW interactions (at certain separation scales) in facilitating and forming colloidal aggregate chains.

To further demonstrate the D–I attraction in a large-scale experiment, the SPS particles were spread at the oil-water interface and were dispersed in pure water (Supplementary Fig. 13). When the particles beneath the oil-water interface slid on a glass substrate and passed by the particles at the interface with long-range separations, some of those underneath the interface attached to the interfacial ones and dragged them through an ordered hexagonal particle array (see the detailed description in Supplementary Note 9). Because the pure water was used as the aqueous subphase, this large-scale experiment clearly demonstrated that the D–I interaction promoted the like-charged colloidal attraction and was responsible for initiating formation of the aggregate dimer.

## Discussion

The D–I interaction, known as the Debye interaction, was investigated based on a colloidal model system. The obtained scaling behavior of the D–I interaction $U_{D–I} \sim r^{-6}$ (due to diffuse layer polarization) at the colloid scale was consistent with the theoretical prediction of the Debye interaction in molecular space. The inverse sixth power arises from the $r^{-3}$ dependence of the magnitude of the induced dipole that is weighted by the $r^{-3}$ dependence of the interaction between the dipole and the induced dipole. Such dipole character could propagate to form colloidal aggregate chains. Furthermore, the D–I interaction initiated like-charged attraction in relatively long-range separations, and the vdW interaction became dominant in short-range separations, leading to aggregate formation. Further investigation will be performed on evaluating the effects of various electrolytes and surfactants on D–I interactions and the micromechanics of colloidal chains. In addition, this study can offer a good model system to quantify the out-of-plane rotation behavior of an interface-trapped particle against pinning of the fluid interface to the particle surface[40,41] as well as the non-equilibrium wetting phenomenon of interfacial particles[42], which have not been investigated thoroughly. Lastly, the D–I interactions may affect the ligand-receptor interactions in biological systems. Similar to the case of the interface-trapped particle and water-immersed particle, a receptor on a cell surface exposed to a medium carries surface charges, and asymmetric counterion distribution can create a permanent dipole on it. When a ligand immersed in the medium approaches the receptor closely, they can be attracted to each other in relatively longer separations than that predicted from the vdW interaction.

## Methods

### Experimental cells

A circular coverslip (Marienfeld, no. 1.5H, Germany) was attached to a glass ring (25 mm diameter × 11 mm height) using UV adhesive (Optical Adhesive 81, Norland, USA). A sessile water drop (~50 μL) containing PS particles was formed on the bottom of the container using a micropipette. Then, 2 mL of n-decane (Acros Organics, USA) was added to cover the sessile drop. The contact angle of the sessile drop in n-decane was approximately ~69°[27]. The particles were washed multiple times by repeating centrifugation and redispersion[28]. Ultrapure water (resistivity ≥18.2 MΩ·cm) was always used as an aqueous phase and for washing the particles and the experimental cell. Any polar impurities in n-decane were removed using aluminum oxide particles (acidic activated, particle size 100–500 μm, Sigma-Aldrich). All fluids, the experimental cell, and particles were freshly prepared each time immediately prior to performing the experiment. Three types of particles, i.e., sulfate-PS (SPS, Invitrogen, USA), amino-PS (APS, Spherotech Inc., USA), and carboxyl-PS (CPS, Invitrogen) were used (Supplementary Table 1). NaCl (Samchun, Korea), LiCl (Sigma-Aldrich), and $CaCl_2$ (Sigma-Aldrich) were used as electrolytes.

### Force measurements

Time-sharing optical laser tweezers[23,24,26,27] were used to optically trap two particles $P_1$ and $P_2$ in water, and then $P_1$ was laterally translated and forced to attach to the oil-water interface. Once $P_1^i$ attached to the interface, its desorption toward the water phase by laser tweezers did not occur in our low laser power condition of $P \approx 7$ mW with corresponding weak trap stiffness ($\kappa_t = 3.91$ pN·μm$^{-1}$) due to the irreversible and strong attachment energy, ~$10^8 \kappa_B T$[19]. As $P_2$ in water was laterally translated toward the interface-attached stationary $P_1^i$ stepwise (Fig. 1b), the interaction force $F_{exp}(r) = \kappa_t \Delta x(r)$ was determined as a function of the center-to-center separation $r$ by measuring the displacement $\Delta x$ of the translated $P_2$ from its corresponding equilibrium positions (Fig. 2a, b). The equilibrium positions of the $P_2$ were determined before or after pair interaction measurements, in which $P_2$ was translated along the same steps without $P_1^i$ (Fig. 2a). The trap stiffness values $\kappa_t$ was measured by using the drag calibration method. We refer the readers to Supplementary Notes 1–3 for the detailed explanation of optical tweezer setup, drag calibration, and image analysis.

Note that the oil–water interface of the sessile drop formed a contact angle of ~69° with the bottom coverslip. The pair interaction measurement was conducted at a distance of ~40–50 μm from the coverslip surface. Given that the height of the sessile drop apex from the coverslip was at a millimeter scale, the angle between the focal plane at which $P_1^i$ was located and the interface could be assumed to be the same as ~69°. Although this geometry might affect the D–I interaction due to the non-parallel approach of $P_2$ to the dipole formed on $P_1^i$, it could be help prevent unquantifiable artifacts resulting from the use of laser tweezers. The highly focused laser beam from the objective lens with $NA = 1.2$ might scatter at the oil–water interface when it trapped a particle near the interface, leading to potential errors in force measurements. However, in the current setup, the objective lens generated a maximum half angle of ~64.5° for the con-like laser beam[43], which was smaller than the contact angle of 69° between the interface and coverslip, thereby eliminating such potential artifacts.

## CGMD simulations

The coarse-grained SPS model was composed of 500 constituent sulfate-styrene monomers[44]. The MARTINI force field was applied, and the bead types for each CG are shown in Supplementary Fig. 5[45]. For simulation of a single SPS particle, 32 SPS chains were constructed in a box of $30 \times 30 \times 30$ nm³. The water environment contained 280 Na⁺ and 280 Cl⁻ ions (i.e., 10 mM NaCl). To create the oil-water interface environment, n-decane molecules were introduced as the oil phase region, and 10 mM NaCl and water molecules were introduced as the aqueous phase region. The equilibrium state of a single SPS particle (which was obtained from the water environment system) was located at the oil–water interface, and the SPS chains in the oil phase were fixed with a harmonic potential. NPT (i.e., isothermal-isobaric) simulations were conducted at 298 K for 100 ns when the single SPS particle was either in the water or at the interface. The ion distribution around the SPS particle was obtained by analyzing the trajectories of the ion CG beads during the last 10 ns. The dipole moment, given by $\mu = \int \mathbf{r} q(\mathbf{r}) d\mathbf{r}$, was estimated for the charged elements (i.e., Na⁺ near the particle and the sulfate groups), where $\mathbf{r}$ is a position vector between the CG beads and $q(\mathbf{r})$ is the charge of the CG bead. In the simulation of the dimer and trimer systems, the initial center-to-center separation between two particles was set to 8 nm. NPT simulations were conducted at 298 K for 1000 ns. All CGMD simulations were conducted at a time step of 10 fs using the GROMACS 2016 package[46]. A velocity rescaling thermostat and Parrinello–Rahman barostat were used for the isothermal–isobaric state. The cutoff radius for the vdW interactions was 1.2 nm. The particle mesh Ewald method was used to treat the Coulomb interactions.

## Data availability

All data are presented in the paper and the Supplementary Information. Any additional information can be obtained from corresponding authors upon request.

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

## Acknowledgements

This work was supported by the National Research Foundation (NRF) of Korea, No. NRF-2021R1A5A6002853 (B.J.P., S.K.K.), NRF-2020R1A2B5B01001949 (B.J.P.), and NRF-2020M3DIA2102915 (S.K.K.). Computational resources were provided by KISTI-HPC (KSC-2021-CRE-00205, S.K.K.). The authors thank Professor Eric M. Furst (University of Delaware) for critical reading of the manuscript.

## Author contributions

B.J.P. and S.K.K. conceived and supervised the project. H.M.L., Y.W.K., and C.R. performed the D-I interaction measurements and the particle attachment probability experiments. K.H.C. obtained the pair interaction data at the interface and the mean dipole strength of interface-trapped particles. E.M.G., Y.C., and S.K.K. performed the coarse-grained MD simulations. All authors contributed to the interpretation of experimental data and read, edited, and commented on this manuscript.

## Competing interests

The authors declare no competing interests.
