## [Peer Review File · Nature Communications]

Direct measurements of the colloidal Debye forceREVIEWER COMMENTS

Reviewer #1 (Remarks to the Author):

In this manuscript, Lee and coworkers present an interesting work on the measurement of Debye forces at the micrometer scale. This is a challenging task, as Debye interactions are usually weaker than other dipole-dipole interactions. While this prevents in practice their direct measurement at the molecular level, upscaling to the micron scale is, in my opinion, a sound strategy to pursue this task. The authors have devised a smart approach to create induced dipoles based on a rearrangement of the diffuse layer of mobile ions around charged colloids. Based on this, they provide convincing experimental evidence of their main result - the measurement of Debye interactions. Interestingly, they also show that the Debye attraction allows building chains of dipoles which can sustain a moderate mechanical stress. In the hope of getting a better idea of the fundamental mechanisms behind their findings, the authors also perform coarse-grained simulations to ascertain the relative importance of micron-scale Debye interactions versus standard van der Waals interactions. I strongly appreciate the experimental findings and their analysis, which in my opinion are worth publishing in Nature Communications. Yet, I am not convinced by the interpretation of the results at the fundamental level, where according to the authors the van der Waals attraction becomes the main player at the smallest distances. As I comment below, I believe that there are some important missing points to support this view, for which the implemented coarse-grained simulations cannot provide answers. Therefore, before recommending publication of this nice work I would need to see a more physically-sound interpretation, or a proper justification of the current one.

As mentioned above, my main concern is in the claim of the authors that van der Waals (vdW) interactions dominate the short-distance energetics. If I understand correctly their point, the role of micron-sized Debye interactions should be to "catch" the other colloids at long range, while adhesion should be maintained by the vdW interactions. What follows is based in assuming that the authors refer to vdW interactions as the "standard" ones, i.e. the sum of Keesom, Debye and London forces at the molecular level and their cumulated effect over the volumes of colloidal particles. To my understanding, in the work there is no actual evidence of the claimed short-range dominance of vdW interactions over micron-sized Debye interactions, and I would rather say that the data support the opposite view (i.e. that micron-scale Debye interactions dominate even at small scales). This is based on the following observations:

a) In the control experiment where both colloids are put in water, putting them in contact does not result in the formation of a stable dimer. If vdW interactions were dominant at short scale, the dimer should be formed and stable. The only way for this to be compatible with the authors' claim is that vdW interactions are different in the control as compared to the water/decane system. Since in the latter the permanent dipole in P1⁺ is formed by the asymmetric ion cloud, I find it hard to believe that this changes the molecular dipoles within the colloids. One might imagine that the presence of decane is somehow responsible, but this would imply that further colloids would not be able to attach, thus impeding the formation of the observed chains. Hence, I believe that this rules out short-range vdW attraction rather than supporting it.

b) The coarse-grained simulations are performed with a force field incapable of polarization. There, vdW interactions are introduced as Lennard-Jones potentials (or similar) and are by construction the same in the pure water and the water/decane setup. If vdW interactions were responsible for holding the dimer together, they should be able to do it for both setups. Have the authors checked that a preformed dimer is stable in simulations in the control setup? The only mention to the matter is "attraction and dimer formation were not observed in the water condition" (page 13, line 252), which suggests that the authors initialized the particles at a certain mutual distance and did not observe dimer formation. What would happen if the dimer was there from the beginning? If the dimer were found to be stable, this

would go against the experimental observations, indicating that the vdW interactions in the simulations are too strong. If not, this would mean that vdW interactions are not strong enough to hold the dimer (in agreement with the experiments), but this would also apply to the water/decane setup, since in the force field the vdW interactions are the same for both setups. Therefore, the observation of dimer stability should be ascribed to other mechanisms, such as the Debye interactions. As a word of caution, it might be anyway dangerous to upscale the results obtained in the simulations (colloids of diameter $D \sim 20$ nm) to the interpretation of experimental results ($D \sim 3$ micron, i.e. 150 times larger). As the authors point out (page 12, line 227), the dipole moment μ scales with the area of the particle surface, i.e. μ is proportional to D^2 . Since μ appears quadratically in the Debye force, this means that the force is upscaled by a factor proportional to D^4 . In contrast, within the DLVO theory the vdW interaction is proportional to D . This means that, for a system of size similar to the experimental setup, the relative intensity of the two forces is expected to change by a factor $\sim 150^3$ in favour of the Debye interactions.

c) If present, according to the standard view of the DLVO theory the vdW interactions would be longer-range than the Debye interactions (the force scaling as $1/r^2$ rather than $1/r^7$), hence they would dominate the force response, which is in contrast with experimental evidence. In this regard, I do not understand the qualitative features of the plot reported in Fig.2e, for which the vdW function goes to zero faster than the Debye interaction. Which formulas did the authors use for this plot?

Coarse-grained simulations can provide useful information on the microscopic origin of the dipoles. However, the data presented for the distributions of the Cl ions in the coarse-grained simulations are too noisy to conclusively claim the skewness of the distribution (Fig.4d and Fig.4e). For instance, the data in the shaded area in Fig.4d for the left region are noisy, and no attempt of error estimation is given. Moreover, what is the relation between Fig.4d and Fig.4e? My understanding would be that the 120 ns point in Fig.4d is obtained as average of the number density points in Fig.4e for each region, but the numbers do not match. Also, I see in Fig.S7 the equivalent of Fig.4e for the water simulations. How does the equivalent of Fig.4d look like? Perhaps the authors should consider running more simulations under the same conditions to collect better statistics.

Apart from these major concerns, I have some minor comments and suggestions which the authors might find worth considering:

1) In Fig.1, the vectors depicting the dipole moments go from the positive to the negative charge. This is in contrast to the common convention, according to which they point from the negative to the positive charge. I suggest to amend the pictures.

2) At page 3, lines 53-54, the authors state that for a permanent dipole with fixed orientation the attraction doubles. This is correct, but a reader not familiar with dipole energetics might get a bit confused. I suggest to write a small section in the Supplementary Information with the main formulas: the dipole-dipole interaction of two parallel dipoles at a generic angle with respect the line joining their centers and the results obtained for i) angle average, ii) joining line parallel to the dipoles. Also, I suggest that the authors write explicitly that ii) is the case at hand, which is reflected in the formula reported at page 7, line 130. Similarly, it would help understanding the origin of the formula at page 8, line 133. From my understanding, this should be obtained by considering a joining line perpendicular to the direction of the dipole, but according to my calculations there should be a 4 at the denominator, rather than a 8. Could the authors countercheck this formula and comment on its derivation?

3) Section "Determination of dimer formation probability": Why are dimers not always formed? The authors suggest that a certain time is necessary for the rearrangement of ions. Yet it seems that the plateau in probability of dimer formation (Fig.3b) is not 1, suggesting that even for long waiting times only a fraction of dimers will be formed. Could the authors comment/speculate on this point?

4) It would be nice to see the goodness of the fit in Fig.4d in log-log scale (considering the

absolute value of the force). The fitting parameters and the range of values in Fig.4d suggest 4-5 orders of magnitudes being spanned.

5) Based on their data, could the authors provide an estimate for the Young modulus of the chain of dipoles? It would be nice to put it in the context of known materials

6) While being a suggestive mechanism, I am unsure that the results of this work would apply to the case of a ligand-receptor interaction (page 18, lines 333-338), as their size (1-10 nanometers) would make the Debye interaction of the kind considered in this paper very weak.

Reviewer #2 (Remarks to the Author):

In their study, Lee et al. investigate dipole-induced dipole (D-I) forces between colloids using optical trapping measurements. A permanent dipole is obtained by pinning a polystyrene particle at an oil/water interface. Colloidal aggregates were observed due to the contribution of van der Waals and DI attractive forces. In addition, the DI forces were shown to propagate to the next particles when creating linear chains. The authors have tested many different parameters including the surface functionalization of the polystyrene particles, the nature and concentration of the electrolyte; and further supported their experimental findings with computer simulations. Altogether, I found this fundamental study very interesting both from the quality of the experiments and the clear presentation of the different results. I do believe it is an important contribution that is addressing many opened questions. I recommend its publication for Nature Communications with minor revisions as I believe some points still need to be addressed by the authors. My comments are the following:

1. Detection of the oil water interface. The authors use bright field microscopy; would it be possible to directly image the oil water interface using for instance DIC microscopy?

2. Position of the particle at interface. The permanent dipole is created by placing the particle at the oil water interface. Do the authors account for the angle formed by the interface as it will affect their measurements if they only manipulate their particles within the imaging plane as shown in Figure 1d? Ideally, one would try to obtain a contact angle close to 90° and to measure close (but far enough) from the coverslip.

3. Do the authors have an idea of the contact angle of the particles at the oil/water interface from the literature? It is certainly an important parameter as it defined the extent of the protrusion of the particle in the oil-phase. How does it compare to their simulations?

4. Dipole-dipole interactions. If the dipole-dipole forces are the product of the asymmetric surface charge dissociation, I would expect them to strongly depend on the ionic strength. Did the authors perform such measurements at different ionic strength or for different electrolytes (i.e. different Debye lengths)?

5. Figure 2d: force measurement. The different contributions to the measured forces are not clearly visible from the linear scale. I would suggest the authors to use a logarithmic scale for the x-axis. In addition, they should detail the different contributions to the total force. What is the expression they used for F_{el} and for F_{vdw} from the DLVO theory? I would suggest the authors to add a paragraph in the supporting with the different parameters and assumptions that were made (surface potential=zeta potential? Effective Hamacker constant, radius, Debye length). An additional force measurement in water at lower ionic strength should be provided to demonstrate the robustness of the force measurements.

6. The colloidal D-I forces seem strongly related to the asymmetric distribution of counterions and to be very much dependent on the particle charge, size and electrolyte concentrations. How much the comparison with the molecular polarizability which is only described in the

paper as a differences of permittivity makes sense?

7. D-I forces. The D-I forces together with vdW forces seem to hold the assembly together. In addition, at lower ionic strength I would still expect repulsive screened Coulomb interactions to take place. Is it the reason why P_f decreases at lower ionic strength? Figure 3c may indicate that the particles hold together due to vdW forces at high ionic strength. Was it possible to measure P_f at higher ionic strength to show that ultimately P_f at interface becomes equal to P_f in water?

8. Following the classical DLVO theory, it could maybe have been interesting to measure the stability of different particles to extract the surface potential from the critical coagulation concentration.

9. The measurements performed with different salt and particles in Figure 3a are very intriguing. Could they further comment on the fact that 10 mM LiCl and 10 mM CaCl₂ provide similar results?

10. Why are the particles in the simulations so rough? In practice, the PS particles should be only surface functionalized and I would expect them to be rather smooth, particularly when their molecular roughness is compared to their diameter.

11. Colloidal chain. Please provide the Euler-Bernoulli equation in the supporting information. As the chain bends, aren't the DI-forces expected to decrease as the permanent dipole maintain is orientation? Could the authors please comment on this? The authors need to define κ_t and δd more precisely in their manuscript. Does F_{bend} can be compared to the total force between two particles? What do they learn from the bending force at rupture?

12. Line 296. "in 10 mM water" should read "in 10 mM NaCl water"

Reviewer #3 (Remarks to the Author):

This manuscript describes measurements and simulations of the interaction between a charged colloidal sphere embedded in an oil-water interface and one or more similar spheres dispersed in the aqueous phase. This system is expected to display Debye attractions between the permanent electric dipole moment of the interfacial particle and induced dipole moments in the neighboring spheres. As the authors point out, colloidal Debye interactions have not been reported previously, nor have their interesting collective effects been described. In addition to reporting optical-tweezer measurements of interparticle forces, the authors use optical tweezers to assemble particles into flexible chains that are anchored by an interfacial particle. The experimental study is supported by molecular dynamics simulations that demonstrate Debye attractions between interfacial and bulk particles under conditions where van der Waals attractions are weak. The subject of this contribution meets the standards of novelty and importance for publication in Nature Communications.

The current manuscript, however, does not provide nearly enough information for the reader to assess whether or not its conclusions are valid. Without this information, I would not recommend publication in any journal. Assuming it can be provided and supports the current manuscript's interpretation, I would be inclined to recommend publication in Nature Communications.

1. The paper should provide details on the instrument. Is it based on a commercial

microscope? If so, which one? If the instrument was custom-built, the text should provide information on its components and its layout. What is the magnification of the imaging system (in micrometers per pixel)? What is the exposure time of the camera? What is the power and wavelength of the trapping laser?

2. How were particle positions measured? Was a standard software package used, or custom software? What algorithm was used to localize the particles' centroids? Does that algorithm account for the overlapping diffraction patterns of particles near contact? This is a particularly important point because such overlaps are known to introduce artifacts into inferred interaction potentials. A standard reference on this point is Baumgartl J, Bechinger C. On the limits of digital video microscopy. *EPL (Europhysics Letters)* 71, 487 (2005).

In explaining the measurement technique, the authors also should specify the precision and accuracy with which their algorithm tracks their particles' three-dimensional positions. Projection errors due to out-of-plane motions are critically important for colloidal force measurements near contact.

3. I presume that interactions were measured by estimating particles' displacements in the potential well of calibrated optical tweezers. If so, this should be explained in the text. How were these calibrations performed? What calibration constants were measured, and what are their uncertainties?

4. If forces were measured by monitoring a particle's displacement in its trap, how large are the trapped particles' thermal fluctuations? In light of these fluctuations, how many particle-separation measurements contribute to each point in the reported force-separation curves?

5. How do we know that the optical tweezer does not affect measured interparticle interactions?

5.a) Trapped particles can be attracted to their neighbors' traps, particularly at small separations. How do we know that this can be ignored? If it cannot be ignored, how was corrected?

5.b) Light scattered by the trapped particle can give rise to inter-particle forces. The need to measure and correct for such effects has been discussed, for example, in Crocker JC, Matteo JA, Dinsmore AD, Yodh AG. Entropic attraction and repulsion in binary colloids probed with a line optical tweezer. *Physical Review Letters* 82, 4352 (1999). Other groups have avoided this problem by turning off the optical tweezers during an interaction measurement, with relevant references including

Crocker JC, Grier DG. Microscopic measurement of the pair interaction potential of charge-stabilized colloid. *Physical Review Letters* 73, 352 (1994); and Sainis SK, Germain V, Dufresne ER. Statistics of particle trajectories at short time intervals reveal fN-scale colloidal forces. *Physical Review Letters* 99, 018303 (2007).

5.c) Light absorbed from the optical tweezer can heat the trapped particle and the surrounding medium. Heating can change charged particles' charges and can induce flows in the surrounding fluid. How do we know that optically-induced heating can be neglected? This also is why it is necessary to specify the wavelength and power of the trapping laser.

5.d) The proximity of the oil-water interface can influence the optical force field through the light that it scatters. How do we know that the trap's position and stiffness were not affected by the proximity of the interface?

6. The data in Fig. 2(d) are supposed to justify the paper's principal conclusions regarding the nature of the interparticle forces. The current presentation, however, is not convincing. Most data points are obtained at such large separations that interparticle forces are negligible. Just a few measurements at very small separations are used to infer the functional form of the interaction force. The manuscript should include a version of this plot that zooms in on the region near contact so that the reader can assess the validity of the

comparison with the Debye force law. A log-log plot would be helpful to assess how well the data are described by power-law scaling and the exponent of that scaling law.

7. The most relevant range of separations for measuring the interparticle force also is the range of separations where the optical tweezer itself can mediate interactions between the particles and where the presence of the interfacial particle can modify the potential energy well of the optical tweezer. It also is the range where diffraction introduces overlap errors in image-based particle tracking. It also is the range where out-of-plane motions can be most significant and also most difficult to measure. If, as I suspect, just a few data points support the interpretation of the experiment, then particular care will be needed to exclude all such artifacts.

REVIEWER COMMENTS

Reviewer #1 (Remarks to the Author):

In this manuscript, Lee and coworkers present an interesting work on the measurement of Debye forces at the micrometer scale. This is a challenging task, as Debye interactions are usually weaker than other dipole-dipole interactions. While this prevents in practice their direct measurement at the molecular level, upscaling to the micron scale is, in my opinion, a sound strategy to pursue this task. The authors have devised a smart approach to create induced dipoles based on a rearrangement of the diffuse layer of mobile ions around charged colloids. Based on this, they provide convincing experimental evidence of their main result - the measurement of Debye interactions. Interestingly, they also show that the Debye attraction allows building chains of dipoles which can sustain a moderate mechanical stress. In the hope of getting a better idea of the fundamental mechanisms behind their findings, the authors also perform coarse-grained simulations to ascertain the relative importance of micron-scale Debye interactions versus standard van der Waals interactions. I strongly appreciate the experimental findings and their analysis, which in my opinion are worth publishing in Nature Communications. Yet, I am not convinced by the interpretation of the results at the fundamental level, where according to the authors the van der Waals attraction becomes the main player at the smallest distances. As I comment below, I believe that there are some important missing points to support this view, for which the implemented coarse-grained simulations cannot provide answers. Therefore, before recommending publication of this nice work I would need to see a more physically-sound interpretation, or a proper justification of the current one.

We thank the reviewer for the positive feedback on our work and for the insightful comments. We agree that a more physically-sound interpretation is necessary to support our claim on the dominance of vdW at short distances. We carefully considered the reviewer's suggestions to improve our interpretation and provide a more thorough justification for our conclusions.

As mentioned above, my main concern is in the claim of the authors that van der Waals (vdW) interactions dominate the short-distance energetics. If I understand correctly their point, the role of micron-sized Debye interactions should be to "catch" the other colloids at long range, while adhesion should be maintained by the vdW interactions. What follows is based in assuming that the authors refer to vdW interactions as the "standard" ones, i.e. the sum of Keesom, Debye and London forces at the molecular level and their cumulated effect over the volumes of colloidal particles. To my understanding, in the work there is no actual evidence of the claimed short-range dominance of vdW interactions over micron-sized Debye interactions, and I would rather say that the data support the opposite view (i.e. that micron-scale Debye interactions dominate even at small scales). This is based on the following observations:

a) In the control experiment where both colloids are put in water, putting them in contact does not result in the formation of a stable dimer. If vdW interactions were dominant at short scale, the dimer should be formed and stable. The only way for this to be compatible with the authors' claim is that vdW interactions are different in the control as compared to the water/decane system. Since in the latter the permanent dipole in P1⁺ is formed by the asymmetric ion cloud,

[Type here]

I find it hard to believe that this changes the molecular dipoles within the colloids. One might imagine that the presence of decane is somehow responsible, but this would imply that further colloids would not be able to attach, thus impeding the formation of the observed chains. Hence, I believe that this rules out short-range vdW attraction rather than supporting it.

As the reviewer pointed out, the colloidal vdW attraction between two particles is expected to be similar, regardless of whether one of them is at the interface or all are in water. Experimental results show that the dimer formation probability in the 10 mM NaCl water-only environment is very low, indicating the presence of an energy barrier between two particles due to the DLVO interaction, which is the sum of double layer repulsion F_{el} and vdW attraction F_{vdW} . It has been reported that in the case of actual particles, both interaction forces can increase due to the surface roughness effect [Suresh and Walz, *J. Colloid Interface Sci.* 183, 199 (1996); Suresh and Walz, *J. Colloid Interface Sci.* 196, 177 (1997); Pantina and Furst, *Langmuir* 20, 3940 (2004)], which could explain the difference between the theoretical calculation of DLVO in Fig. 2d,e and the experimental results.

On the other hand, when one particle is at the oil–water interface, the ion rearrangement around the interface-trapped particle and a water-immersed particle nearby can cause a decrease in double layer repulsion and the generation of D–I attraction. This means that the energy barrier between the two particles can be reduced or eliminated. Thus, they can approach spontaneously to a vdW dominant region. However, when the two particles are nearly in contact, there may not be enough ions in the diffuse layer between the two particles to generate an induced dipole, so the influence of the D–I attraction generated by the ion rearrangement would be reduced.

One of the experimental pieces of evidence supporting the claim that vdW affects the formation of dimers and colloidal chains near contact is that the mechanical properties of the aggregate chain in Fig. 6d-f are consistent with the typical behavior of colloidal aggregate chains formed by vdW, such as resistance to bending moment and small-scale rearrangement/rupture between particles due to critical shear force [Pantina and Furst, *Phys. Rev. Lett.* 94, 13801 (2005)]. In particular, the rupture phenomenon observed in the colloidal chain in Fig. S12 confirms that the particles were trapped in a shallow secondary energy minimum due to the vdW force [Pantina and Furst, *Langmuir* 20, 3940, 2004].

In summary, in the decane/water environment, the D–I attraction is believed to primarily act at relatively long distances, allowing the particles to spontaneously approach each other without experiencing an energy barrier. When they are nearly in contact, the sparse ion distribution between the particles likely leads to the reduction in the D–I interaction, and adhesion due to vdW occurs dominantly in such a separation range. We clarified these points in the revised manuscript.

Before: We speculated that the ion distribution between two particles can be sparse when they are sufficiently close. As a result, the diffuse layer polarization around P_2 is reduced, and the two particles can attach to each other under the influence of vdW.

[Type here]

After (p12): The simulations may also provide insight into whether the D–I interaction persists as the particles get closer or whether it disappears at a certain distance due to sparse ion distribution between them. If the two particles are nearly in contact, the diffuse layer polarization around P_2 may be reduced, and they may primarily attach to each other under the influence of vdW.

Before: Then, the ion number density between the two particle regions decreased with time, suggesting that the ions diffused out as the particles approached each other.

After (p13): Then, the Cl^- ion number density between the two particle regions decreased with time, indicating that the ions diffused out as the particles approached each other. The sparse ion distribution between the particles could have contributed to the reduction in the D–I attraction.

Before: Local rearrangements between the particles and chain rupture were observed when a critical bending moment was applied to a colloidal chain in another bending experiment (pentamer in Fig. S9). Overall, the results consistently demonstrated the separate roles of D–I and vdW interactions (at certain separation scales) in facilitating and forming colloidal aggregate chains.

After (p17): Local rearrangements between the particles and chain rupture were observed when a critical bending moment was applied to a colloidal chain in another bending experiment (pentamer in Fig. S12). These results indicated that the particles was in a shallow secondary energy minimum formed by the colloidal vdW force that could be detached by optical tweezers.³² Overall, the results consistently demonstrated the separate roles of D–I and vdW interactions (at certain separation scales) in facilitating and forming colloidal aggregate chains.

b-1) The coarse-grained simulations are performed with a force field incapable of polarization. There, vdW interactions are introduced as Lennard-Jones potentials (or similar) and are by construction the same in the pure water and the water/decane setup. If vdW interactions were responsible for holding the dimer together, they should be able to do it for both setups. Have the authors checked that a preformed dimer is stable in simulations in the control setup? The only mention to the matter is "attraction and dimer formation were not observed in the water condition" (page 13, line 252), which suggests that the authors initialized the particles at a certain mutual distance and did not observe dimer formation. What would happen if the dimer was there from the beginning? If the dimer were found to be stable, this would go against the experimental observations, indicating that the vdW interactions in the simulations are too strong. If not, this would mean that vdW interactions are not strong enough to hold the dimer (in agreement with the experiments), but this would also apply to the water/decane setup, since in the force field the vdW interactions are the same for both setups. Therefore, the observation of dimer stability should be ascribed to other mechanisms, such as the Debye interactions.

We thank the reviewer for the valuable comment. We considered the interactions between charged particles by introducing charges to beads, where it was necessary to mimic the

[Type here]

experimental system (Fig. S5), so that the polarization could be observed by collective charge differences in our system. Also, as with our response to the comment (a), vdW acts similarly in both water-only and decane/water setups. In the water-only condition, the experimental result of low dimer formation probability (Fig. 3c) when two particles were brought as close as possible using optical tweezers can be interpreted as the presence of an energy barrier between them. Conversely, the higher probability in the decane/water setup can be attributed to ion rearrangement, leading to a decrease in double layer repulsion and an increase in D–I attraction, causing particles to spontaneously approach the secondary energy minimum.

As the reviewer mentioned, a preformed dimer remained stable in CGMD simulations when the two particles were initially brought as close as possible (Fig. S10), indicating that the initial state of the simulation had already overcome the energy barrier between them and corresponded to the state where the particles had fallen in the secondary energy minimum. The reason for using the initial position with a certain separated distance between two water-immersed particles in the original CGMD simulation in Fig. S9 was to confirm that they did not spontaneously approach each other due to the absence of the D–I interaction, which was different from the behavior observed in the decane/water environment (Fig. 4). We revised manuscript to clarify this point.

Before: The simulation results showed overall consistency with the experimental observations and the theoretical predictions from the DLVO interactions, as discussed in Fig. 2d,e. The D-I interaction initiated attraction in relatively long-range separations, and vdW attraction became dominant in relatively short-range separations. When two SPS particles were completely immersed in water, they did not show such an asymmetric anion distribution (Fig. S7). Consequently, attraction and dimer formation were not observed in the water condition, demonstrating the crucial role of the D-I interaction in initiating the attraction over long-range separations at the oil-water interface.

After (P13): The simulation results demonstrated that the D–I interaction played a critical role in initiating attraction in long-range separations, whereas vdW attraction became dominant in short-range separations, consistent with the experimental observations and the theoretical predictions from the DLVO interactions discussed in Fig. 2d,e. To further investigate the importance of the D–I interaction, a CGMD simulation was conducted for two water-immersed particles using a similar initial condition to that used in Fig. 4a, where the particles were initially separated with a finite surface-to-surface distance. The results showed that the asymmetric anion distribution was not observed (Fig. S9), and thus, attraction and dimer formation did not occur in the water-only condition, highlighting the crucial role of the D–I interaction in initiating attraction over long-range separations at the oil–water environment. In contrast, when the two particles were initially brought as close as possible, the preformed dimer remained stable (Fig. S10), indicating that the initial state of the simulation had already fallen into the secondary energy minimum.

Added Fig. S10 (p18, SI):

[Type here]

Fig. S10. CGMD simulation in water when two particles are initially brought as close as possible. Yellow beads represent SPS. Na^+ , Cl^- , and water molecules were omitted for clarity.

(b-2) As a word of caution, it might be anyway dangerous to upscale the results obtained in the simulations (colloids of diameter $D \sim 20$ nm) to the interpretation of experimental results ($D \sim 3$ micron, i.e. 150 times larger). As the authors point out (page 12, line 227), the dipole moment μ scales with the area of the particle surface, i.e. μ is proportional to D^2 . Since μ appears quadratically in the Debye force, this means that the force is upscaled by a factor proportional to D^4 . In contrast, within the DLVO theory the vdW interaction is proportional to D . This means that, for a system of size similar to the experimental setup, the relative intensity of the two forces is expected to change by a factor $\sim 150^3$ in favour of the Debye interactions.

We appreciate the insightful comment from the reviewer. As the reviewer mentioned, we understand the concern that the relative contribution of D–I attraction could become higher when the simulation results are scaled up to the experimental system. This implies that the dipole strength of the P_1^i particle used in the CGMD simulation may need to be smaller than that used in the current study. However, the key results obtained from the simulation are as follows:

- Verification of dipole formation due to the asymmetric ion distribution of the P_1^i particle attached to the interface.
- Confirmation of induced ion rearrangement around the P_2 particle dispersed in water by the dipole of the P_1^i particle.
- Confirmation of relatively long-range attraction (i.e., D–I attraction) between the two particles.
- Confirmation of the presence of D–I attraction through the difference in behavior between two particles in water-only and decane/water environments.

These results would not significantly change even if the dipole strength was changed in the simulation condition.

Furthermore, as mentioned in our responses to comments (a) and (b-1), the negligible contribution of vdW interaction to relatively long-range attraction compared to D–I attraction in the experimental system is supported by the following experimental results: (1) the absence of long-range attraction observed in the water-only environment (Fig. S3), and (2) the significantly larger and longer-range attraction measured experimentally in the decane/water

[Type here]

environment compared to vdW interactions predicted by the DLVO theory (Fig. 2). We have incorporated these points in the revised manuscript.

Before: We experimentally proved that the interfacial particle and the water-immersed particle exhibited D-I interactions when in close proximity to each other. We were still uncertain about whether the D-I interaction is maintained as the two particles get closer or whether the vdW contribution becomes dominant at a certain distance. This question previously arose from the DLVO prediction and the experimentally measured force in Fig. 2d,e. We speculated that the ion distribution between two particles can be sparse when they are sufficiently close. As a result, the diffuse layer polarization around P_2 is reduced, and the two particles can attach to each other under the influence of vdW. To support this hypothesis, we performed coarse-grained molecular dynamic (CGMD) simulations (Fig. S4). Fundamentally, we were eager to find rational answers to the following questions. First, does a dipole form around P_1^i at the oil-water interface? Second, does ion rearrangement or diffuse layer polarization occur around the water-immersed P_2 due to the P_1^i dipole? Third, can an attractive force (i.e., D-I interaction) between the two particles arise in the separation range where the vdW contribution is relatively weak? Fourth, can the effects of D-I and vdW interactions be separated based on interparticle distances?

After (p12): We experimentally demonstrated the presence of the D-I interaction between the interfacial particle and the water-immersed particle at relatively long-range separations. To further support this, we performed coarse-grained molecular dynamic (CGMD) simulations (Fig. S5) to answer several questions. These questions included whether a dipole forms around P_1^i at the oil-water interface, whether ion rearrangement or diffuse layer polarization occurs around the water-immersed P_2 due to the P_1^i dipole, whether this ion rearrangement can attract the two particles to each other, and whether the presence of D-I interaction can be confirmed by comparing the particles' behavior in water-only and decane/water environments. The simulations may also provide insight into whether the D-I interaction persists as the particles get closer or whether it disappears at a certain distance due to sparse ion distribution between them. If the two particles are nearly in contact, the diffuse layer polarization around P_2 may be reduced, and they may primarily attach to each other under the influence of vdW.

Added paragraph (p16, SI): The CGMD simulation aimed to confirm several aspects: firstly, the dipole formation due to the asymmetric ion distribution around the P_1^i particle attached to the interface; secondly, the induced ion rearrangement around the P_2 particle dispersed in water by the dipole of the P_1^i particle; thirdly, the existence of relatively long-range attraction (i.e., D-I attraction), and fourthly, the confirmation of the difference in behavior between two particles in water-only and decane/water environments. It is worth noting that scaling up the simulation results to the experimental scale may increase the relative contribution of D-I attraction compared to vdW attraction, since the D-I potential is proportional to $\sim d^4$ and the vdW potential is proportional to $\sim d$, where d is the particle diameter. Nonetheless, interpreting the above items based on the results of the CGMD simulation would be reasonable.

c) If present, according to the standard view of the DLVO theory the vdW interactions would be longer-range than the Debye interactions (the force scaling as $1/r^2$ rather than $1/r^7$), hence

[Type here]

they would dominate the force response, which is in contrast with experimental evidence. In this regard, I do not understand the qualitative features of the plot reported in Fig. 2e, for which the vdW function goes to zero faster than the Debye interaction. Which formulas did the authors use for this plot?

The reason why the trend of F_{vdW} decreasing faster as r/d approaches 1 in Fig. 2e is because when r approaches d , the vdW potential $U_{vdW} \sim -(r-d)^{-1}$ becomes negative infinity. Thus, the D-I potential $U_{D-I} \sim -r^{-6}$ decays less steeply than U_{vdW} near the region of $r \approx d$. The corresponding forces should behave similarly. The equations related to the DLVO theory have been added to the revised SI.

Added paragraph (p12, SI): The DLVO interaction theory provides a framework for describing colloidal interactions in an aqueous phase.^{12, 20} For two spherical colloids with an equal diameter d , the vdW interaction can be expressed as $U_{vdW} = -\frac{A_H d}{24(r-d)}$, where $A_H = 1.4 \times 10^{-20} J$ is the Hamaker constant between two PS particles interacting across water.¹² The double layer interaction between two colloidal spheres is given by $U_{el} = 32 \times 10^3 \pi d k_B T I N_A \kappa^{-2} Y_0^2 \exp(-\kappa(r-d))$ for $\kappa \left(\frac{d}{2}\right) > 10$, where $I = 10$ mM is the ionic strength, N_A is Avogadro's number, $Y_0 = \tanh \frac{e\psi}{4k_B T}$ is the Gouy-Chapman parameter, $\psi = -57.5$ mV is the particle zeta-potential, e is the elementary charge, and $\kappa = \sqrt{\frac{2000e^2 I N_A}{\epsilon_W \epsilon_0 k_B T}}$ is the inverse Debye screening length.^{20, 21} Note that the U_{el} equation agrees well with the numerical solution of the non-linear Poisson-Boltzmann equation, even at small interparticle separations ($\kappa(r-d) < 1$).²² The corresponding force can be calculated numerically as the derivative of the potential energy with respect to interparticle separation, i.e., $F = -\frac{dU}{dr}$. The DLVO interaction forces between two particles, where one particle is attached to the oil-water interface and the other is immersed in water (Fig. 2d and 2e), were also estimated using the above equations.

Coarse-grained simulations can provide useful information on the microscopic origin of the dipoles. However, the data presented for the distributions of the Cl ions in the coarse-grained simulations are too noisy to conclusively claim the skewness of the distribution (Fig. 4d and Fig. 4e). For instance, the data in the shaded area in Fig. 4d for the left region are noisy, and no attempt of error estimation is given. Moreover, what is the relation between Fig. 4d and Fig. 4e? My understanding would be that the 120 ns point in Fig. 4d is obtained as average of the number density points in Fig. 4e for each region, but the numbers do not match. Also, I see in Fig. S7 the equivalent of Fig. 4e for the water simulations. How does the equivalent of Fig. 4d look like? Perhaps the authors should consider running more simulations under the same conditions to collect better statistics.

We thank the reviewer for the valuable comment, and we apologize for any misunderstandings that may have arisen due to the noisy data and unclear presentation of the number density in Fig. 4d and 4e. To address the noise issue, we conducted two additional independent

[Type here]

simulations and calculated the average. Furthermore, we would like to clarify that Fig. 4d depicts the distribution of chlorine ions between two SPS particles over time, while Fig. 4e shows the ion distribution between them at 120 ns. The discrepancy between the previous figures was due to the use of different sets of volumes in each analysis. Therefore, we used the same volume and added schematics to illustrate the analyzed region as insets in the revised Fig. 4d and 4e.

Revised Fig. 4d and 4e:

Fig. 4. Simulation of $P_1^i - P_2^i$ dimer formation. **d**, Number density of Cl⁻ ions in the left and right regions between the two SPS particles. **e**, Number density of Cl⁻ and Na⁺ between the two particles at $t = 120$ ns. The black dashed line represents the center of the two particles, and the pink area shows a Cl⁻ rich region. The error bars in panels **d** and **e** indicate three independent simulation runs, the insets represent the analyzed regions.

As suggested by the reviewer, a similar analysis was conducted in a water-only environment and added to the revised SI. The distribution of Cl⁻ ion was found to be similar in the left and right regions from the center between two SPS particles under the water-only condition.

Revised Fig. S9:

Fig. S9. CGMD simulation in water when two particles are initially separated with a finite surface-to-surface distance. **a**, Initial and final configurations of two water-immersed SPS particles. Yellow, red, and blue represent SPS, Na⁺, and Cl⁻, respectively. Water molecules

[Type here]

were omitted for clarity. **b**, Number density of Cl^- and Na^+ between the two particles over the simulation time. The black dashed line represents the center of the two particles. **c**, Number density of Cl^- in the left and right regions between the two particles.

Apart from these major concerns, I have some minor comments and suggestions which the authors might find worth considering:

1) In Fig.1, the vectors depicting the dipole moments go from the positive to the negative charge. This is in contrast to the common convention, according to which they point from the negative to the positive charge. I suggest to amend the pictures.

In chemistry, it is conventionally accepted that the vectors representing the dipole moment go from positive to negative. However, in physics, the opposite direction is usually employed. As shown in Fig. 1a, the former notation is commonly used for the dipole vector due to differences in electronegativity. For consistency, we have decided to maintain the current notation in Fig. 1b as well.

2) At page 3, lines 53-54, the authors state that for a permanent dipole with fixed orientation the attraction doubles. This is correct, but a reader not familiar with dipole energetics might get a bit confused. I suggest to write a small section in the Supplementary Information with the main formulas: the dipole-dipole interaction of two parallel dipoles at a generic angle with respect the line joining their centers and the results obtained for i) angle average, ii) joining line parallel to the dipoles. Also, I suggest that the authors write explicitly that ii) is the case at hand, which is reflected in the formula reported at page 7, line 130.

As the reviewer suggested, we added a new section in SI including the main formulas for the dipole-dipole interaction and dipole-induced dipole interaction. Additionally, we revised the formula presented on page 7, line 130 of the original manuscript. The revision changed the equation from $F_0 = -\frac{3\mu_1^2\alpha'_2}{\pi\epsilon_0\epsilon_w d^7}$ to $F_0 = -\frac{3\mu_1^2\alpha'_2}{2\pi\epsilon_0\epsilon_w d^7}$. This change was made because the colloidal Debye interaction is unlikely strong enough to align the dipole and induce dipole parallelly. Therefore, it is more appropriate to use the angle average of $\cos^2 \theta = \frac{1}{3}$, rather than assuming a fixed parallel orientation with $\theta = 0^\circ$. Based on the revised equation and new D-I force measurements, we found excellent agreement between the experimentally obtained polarizability volume $\alpha'_{2,exp} \approx 6.29$ and the theoretical prediction $\alpha'_{eff} \approx 6.47 \text{ } \mu\text{m}^3$.

Added paragraphs (p5, SI):

General formulas for interactions between molecules.^{11, 12}

When two permanent point dipoles μ_1 and μ_2 are fixed at a mutual orientation angle of θ and separated by a distance r in a medium, their dipole-dipole (D–D) potential energy can be expressed as $U_{D-D, fixed} = \frac{\mu_1\mu_2 f(\theta)}{4\pi\epsilon_0\epsilon_r} r^{-3}$, where ϵ_0 is the vacuum permittivity, ϵ_r is the dielectric constant of the medium, and $f(\theta) = 1 - 3\cos^2 \theta$. The r^{-3} dependence arises from the field strength of the two point dipoles (r^{-1} dependence) and the magnitude of each dipole decreasing as r increases (r^{-2} dependence). For two freely rotatable dipoles, their relative orientation is constrained by the interaction strength (r^{-3} dependence). The Keesom

[Type here]

interaction, which is the first contribution to the van der Waals (vdW) interaction, is always attractive and can be described by combining the Boltzmann equation (r^{-3} dependence) with $U_{D-D, fixed}$: $U_{D-D, free} = -\frac{2}{3k_B T} \frac{\mu_1^2 \mu_2^2}{(4\pi\epsilon_0\epsilon_r)^2} r^{-6}$, where k_B is the Boltzmann constant and T is the temperature.

When nonpolar molecules are exposed to an electric field, their electronic distribution and nuclear positions are distorted, inducing a temporary dipole moment. For moderate field strengths E , the magnitude of the induced dipole moment μ^* is linearly proportional to E and can be described by $\mu^* = \alpha E$, where α represents the molecular polarizability. The polarizability volume α' can be expressed as $\alpha' = \frac{\alpha}{4\pi\epsilon_0\epsilon_r} = \frac{\mu^*}{4\pi\epsilon_0\epsilon_r E}$. A permanent dipole μ_1 can induce a dipole moment μ_2^* in a nonpolar, polarizable molecule. The interaction between the dipole and the induced dipole (D-I) is attractive and is referred to as the Debye interaction, which is the second contribution to the vdW interaction. The Debye interaction potential is given by $U_{D-I} = -\frac{1}{2}\alpha_2 E^2$, where $E = \frac{\mu_1(1+3\cos^2\theta)^{\frac{1}{2}}}{4\pi\epsilon_0\epsilon_r r^3}$ is the electric field generated by μ_1 . When the Debye interaction is not strong enough to mutually orient the molecules, the angle average of $\cos^2\theta = \frac{1}{3}$ is used to describe U_{D-I} , which is given by $U_{D-I} = -\frac{\mu_1^2\alpha_2}{(4\pi\epsilon_0\epsilon_r)^2} r^{-6} = -\frac{\mu_1^2\alpha_2'}{4\pi\epsilon_0\epsilon_r} r^{-6}$. The inverse sixth power arises from the r^{-3} dependence of the magnitude of the induced dipole, which is weighted by the r^{-3} dependence of the interaction between the dipole and the induced dipole. The corresponding D-I interaction force is given by $F_{D-I} = -\frac{dU_{D-I}}{dr} = F_0 \left(\frac{d}{r}\right)^7$, where $F_0 = -\frac{3\mu_1^2\alpha_2'}{2\pi\epsilon_0\epsilon_r d^7}$. When the two molecules are mutually oriented with $\theta = 0^\circ$ and thus $\cos^2\theta = 1$, the attraction is doubled, i.e., $U_{D-I}(\theta = 0^\circ) = -\frac{\mu_1^2\alpha_2'}{2\pi\epsilon_0\epsilon_r} r^{-6}$.

Before: It is given by $U_{Debye} = -\frac{\mu_1^2\alpha_2'}{4\pi\epsilon_0\epsilon_r} r^{-6}$, where ϵ_0 is the vacuum permittivity, ϵ_r is the dielectric constant of a medium, α_2' is the polarizability volume, and r is the separation between the two molecules.^{6,7} If the permanent dipole has a fixed orientation, the attraction doubles, i.e., $U_{D-I} = 2U_{Debye}$. In general, the α_2' on a molecule with radius R is approximately equal to the molecular volume, $\alpha_2' \sim R^3$.⁷

After (p3): It is given by $U_{D-I} = U_{Debye} = -\frac{\mu_1^2\alpha_2'}{4\pi\epsilon_0\epsilon_r} r^{-6}$, where ϵ_0 is the vacuum permittivity, ϵ_r is the dielectric constant of a medium, α_2' is the polarizability volume, and r is the separation between the two interacting molecules (a detailed description was provided in Supplementary Information, SI).^{6,7} Typically, the polarizability volume of a molecule α_2' is approximately equal to its molecular volume.⁷

Before: The D-I interaction force was given by $F_{D-I} = -\frac{dU_{D-I}}{dr} = F_0 \left(\frac{d}{r}\right)^7$, where $F_0 = -\frac{3\mu_1^2\alpha_2'}{\pi\epsilon_0\epsilon_r d^7}$. The permanent dipole moment μ_1 for the interface-trapped particles can be obtained using the self-potential method²⁴ by measuring the dipole-dipole pair interaction

[Type here]

forces between i and j particles at a planar oil–water interface $F_{D-D} = \frac{3\mu_i\mu_j}{8\pi\epsilon_0\epsilon_{oil}d^4} \left(\frac{d}{r}\right)^4$ (Fig. S1a). The mean value for 18 particles was $\langle\mu_1\rangle \times 10^4 = 4.5 \pm 2.0$ pC· μm at the same fluid condition (Fig. S1b and S1c). Consequently, the polarizability volume was $\alpha'_2 = \alpha'_{2,exp} \approx 1.49$ μm^3 at $F_0 = -0.208$ pN, which was in order-of-magnitude agreement with the theoretical prediction of the molecular polarizability volume in vacuum, $\alpha'_{2,theory} \sim \left(\frac{d}{2}\right)^3 \approx 3.2$ μm^3 . Notably, a dielectric sphere in a solvent medium with dielectric constants ϵ_p and ϵ_w can be polarized by an electric field, and the effective polarizability volume α'_{eff} can be reduced by a factor of $\left|\frac{\epsilon_p - \epsilon_w}{\epsilon_p + 2\epsilon_w}\right|$,⁷ resulting in $\alpha'_{eff} \approx 1.54$ μm^3 , which shows excellent agreement with the experimental value $\alpha'_{2,exp}$.

After (p7): The D–I interaction force can be expressed as $F_{D-I} = -\frac{dU_{D-I}}{dr} = F_0 \left(\frac{d}{r}\right)^7$, where $F_0 = -\frac{3\mu_1^2\alpha'_2}{2\pi\epsilon_0\epsilon_w d^7}$ and ϵ_w represents the water dielectric constant. The permanent dipole moment μ_1 for the interface-trapped particles can be obtained using the self-potential method²³ by measuring the dipole–dipole pair interaction forces between i and j particles at a planar oil–water interface $F_{D-D} = \frac{3\mu_i\mu_j}{8\pi\epsilon_0\epsilon_{oil}d^4} \left(\frac{d}{r}\right)^4$, where ϵ_{oil} is the decane dielectric constant (Fig. S2a).^{20, 29} The mean value for 18 particles was found to be $\langle\mu_1\rangle \times 10^4 = 4.5 \pm 2.0$ pC· μm at the same fluid condition (Fig. S2b and S2c). The polarizability volume was then calculated as $\alpha'_2 = -\frac{2\pi\epsilon_0\epsilon_w d^7 F_0}{3\mu_1^2} = \alpha'_{2,exp} \approx 6.29$ μm^3 at $F_0 = -0.44$ pN, which was in order-of-magnitude agreement with the theoretical prediction of the molecular polarizability volume in vacuum, $\alpha'_{2,theory} \sim \frac{4\pi}{3} \left(\frac{d}{2}\right)^3 \approx 13.6$ μm^3 . Additionally, a dielectric sphere in a solvent medium with dielectric constants ϵ_p and ϵ_w can be polarized by an electric field, and the effective polarizability volume α'_{eff} can be reduced by a factor of $\left|\frac{\epsilon_p - \epsilon_w}{\epsilon_p + 2\epsilon_w}\right|$,⁷ resulting in $\alpha'_{eff} \approx 6.47$ μm^3 , which shows excellent agreement with the experimental value $\alpha'_{2,exp}$.

Similarly, it would help understanding the origin of the formula at page 8, line 133. From my understanding, this should be obtained by considering a joining line perpendicular to the direction of the dipole, but according to my calculations there should be a 4 at the denominator, rather than a 8. Could the authors countercheck this formula and comment on its derivation?

Regarding the expressions of the dipole-dipole interaction potential $U_{D-D} = \frac{\mu_i\mu_j}{8\pi\epsilon_0\epsilon_{oil}r^3}$ and the corresponding force $F_{D-D} = \frac{3\mu_i\mu_j}{8\pi\epsilon_0\epsilon_{oil}d^4} \left(\frac{d}{r}\right)^4$, we added citations [Hurd, A. J. Journal of Physics A: Mathematical and General 45, L1055 (1985); Oettel, M. and Dietrich, S. Langmuir 24, 1425 (2008)] in the revised manuscript. As pointed out by the reviewer, this equation differs by a factor of 0.5 and the use of the oil dielectric constant ϵ_{oil} compared to the standard version of the dipole-dipole interaction. These differences reflect that when two point dipoles located at an oil-water interface are aligned perpendicularly to the interface, the resulting dipole-dipole

[Type here]

repulsion is mainly originated by the electric field overlapping in the oil phase, rather than in the aqueous phase. We added a paragraph in the revised SI to clarify this point.

Added paragraph (p7, SI):

Dipole strength of an interface-trapped particle.

When a charged colloidal particle is attached to an oil–water interface, an electric dipole can be formed due to the asymmetric surface charge dissociation across the interface.¹³⁻¹⁵ The D–D interaction potential between two particles at the interface and the corresponding force are given by $U_{D-D} = \frac{\mu_i \mu_j}{8\pi \epsilon_0 \epsilon_{oil} r^3}$ and $F_{D-D} = -\frac{dU_{D-D}}{dr} = \frac{3\mu_i \mu_j}{8\pi \epsilon_0 \epsilon_{oil} d^4} \left(\frac{d}{r}\right)^4$, respectively, assuming that each dipole is perpendicular to the interface.¹⁶ Note that this expression differs by a factor of 0.5 and the use of the oil dielectric constant ϵ_{oil} compared to the standard version of the D–D interaction. These differences reflect that when two point dipoles, μ_i and μ_j , located at the interface are aligned perpendicularly to the interface, the resulting D–D interaction is mainly originated by the electric field overlapping in the oil phase, rather than in the aqueous phase.

3) Section "Determination of dimer formation probability": Why are dimers not always formed? The authors suggest that a certain time is necessary for the rearrangement of ions. Yet it seems that the plateau in probability of dimer formation (Fig.3b) is not 1, suggesting that even for long waiting times only a fraction of dimers will be formed. Could the authors comment/speculate on this point?

We believe that the reason for not achieving 100% dimer formation probability for all particle pairs is due to the heterogeneity of the particles. Interaction heterogeneity of colloidal particles has been reported in many previous studies [e.g., Park et al. *Langmuir* **24**, 1686-1694 (2008); Choi K. H., et al. *ACS Appl. Polym. Mater.* **2**, 1304-1311 (2020); Park et al. *Soft Matter* **6**, 5327-5333 (2010)], and one of the main reasons for this is known to be the surface charge nonuniformity of the particles [Feick and Velegol, *Langmuir* 3454, 18 (2002); Feick et al., *Langmuir* 3090, 20, 2004]. We added this point to the revised manuscript.

Before: Higher values of t_h increased the probability of attraction, which suggests that a certain amount of time might be required to allow ion rearrangement needed to generate D-I attraction.

After (p10): Higher values of t_h increased the probability of attraction, which suggests that a certain amount of time might be required to allow ion rearrangement needed to generate D–I attraction. However, the cumulative P_f value did not reach 1, possibly due to the interaction heterogeneity^{22, 24, 28} resulting from the surface charge nonuniformity of the particles.^{33, 34}

4) It would be nice to see the goodness of the fit in Fig.4d in log-log scale (considering the absolute value of the force). The fitting parameters and the range of values in Fig.4d suggest 4-5 orders of magnitudes being spanned.

[Type here]

It appears that the reviewer was referring to Fig. 2d instead of Fig. 4d. We conducted measurements of the D–I interaction force for a larger number of particle pairs and presented them in the revised Fig. 2d with a log-scale x-axis. Each force profile was fitted using the formula $F = F_0 \left(\frac{d}{r}\right)^b$, and the resulting fitting parameters, the force magnitude F_0 and the power-law exponent b , were histogrammed and added as insets in Fig. 2d.

Revised Fig. 2d:

Fig. 2. Direct measurements of the D–I interaction force. d, Measured D–I interaction force profile (F_{D-I}) and comparison with the DLVO forces (F_{el} and F_{vdW}). The water phase contains 10 mM NaCl. The x-axis of the graph is on a logarithmic scale. The red solid line represents a fitted curve that uses the mean values of the two fitting parameters, $\langle F_0 \rangle = -0.44$ and $\langle b \rangle = 7.18$. The black dotted lines indicate the guideline for $F \sim r^{-7}$. The insets show histograms of the values of the two fitting parameters for 72 pairs. The force profiles display before the paired particles come into contact.

Before: We measured the forces for ~ 30 pairs due to the interaction heterogeneity present when measuring the colloidal interaction forces.^{22, 24, 28} Among these pairs, the force profiles (F_{D-I}) of seven with considerable attraction magnitude are shown in Fig. 2d. The other pairs did not show measurable attractive forces. The measured force was fitted with $F_{fit} = F_0 \left(\frac{d}{r}\right)^b$. Two-parameter fitting (inset in Fig. 2d) resulted in average values of $\langle F_0 \rangle = -0.208 \pm 0.133$ pN and $\langle b \rangle = 6.913 \pm 0.854$ with $\langle \chi^2 \rangle = 1.14 \times 10^{-3}$, and one-parameter fitting with a fixed power law exponent of $b = 7$ resulted in values of $\langle F_0 \rangle = -0.168 \pm 0.097$ pN with $\langle \chi^2 \rangle = 1.19 \times 10^{-3}$.

After (p7): We measured the forces for 290 pairs due to the interaction heterogeneity present when measuring the colloidal interaction forces.^{22, 23, 28} Among these pairs, the 72 force profiles

[Type here]

(F_{D-I}) with considerable attraction magnitude are shown in Fig. 2d. The other pairs did not show measurable attractive forces. The measured force was fitted with $F_{fit} = F_0 \left(\frac{d}{r}\right)^b$. Two-parameter fitting (insets in Fig. 2d) resulted in average values of $\langle F_0 \rangle = -0.44 \pm 0.17$ pN and $\langle b \rangle = 7.18 \pm 1.22$ with $\langle \chi^2 \rangle = 9.87 \times 10^{-3}$.

5) Based on their data, could the authors provide an estimate for the Young modulus of the chain of dipoles? It would be nice to put it in the context of known materials

As per the reviewer's suggestion, we estimated the Young's modulus of the PS chain. The detailed procedure and description have been included in the revised SI.

Before: In addition, the corresponding force profiles during the bending and relaxing events were similar (Fig. 6f), indicating no small-scale rearrangements between the particles due to a critical bending moment.

After (p17): In addition, the corresponding force could be obtained by using $F_{bend}(\delta) = \kappa_t \Delta d$, where δ is the chain deflection and Δd is the displacement of the P_7 particle from its equilibrium position. The resulting profiles during the bending and relaxing events were similar (Fig. 6f), indicating no small-scale rearrangements between the particles due to a critical bending moment. The bending rigidity of the colloidal chain was estimated to be $\kappa_{chain} \approx 0.41$ pN/ μm by linear regression of the force profile in Fig. 6f. Using the Johnson–Kendall–Roberts (JKR) theory for particle adhesion, we estimated the Young's modulus of the PS chain to be ~ 0.05 GPa, which was comparable to those of e.g., polystyrene foam (~ 0.005 GPa) and low density polyethylene (~ 0.2 GPa) (more details can be found in SI).^{37, 38}

Added paragraph (p9, SI):

Bending experiment of colloidal chains.^{17, 18}

The deflection y of a colloidal chain with a length L under an applied load F can be described by the Euler-Bernoulli beam equation, $y(x) = \frac{F}{6EI} (3Lx^2 - x^3)$, where E is the Young's modulus and I is the area moment of inertia. The bending rigidity of a single-bonded colloidal linear chain is expressed as $\kappa_{chain} = \frac{F_{bend}}{\delta}$. In Fig. 6f, the linear regression of the force profile resulted in $\kappa_{chain} \approx 0.41$ pN/ μm . The single-bond rigidity κ_0 can be defined as $\kappa_0 = \kappa_{chain} \left(\frac{s}{R}\right)^3 = \frac{3\pi a_c^4 E_{chain}}{4R^3}$, where s is the chain contour length, R is the particle radius, E_{chain} is the Young's modulus of colloidal chain, and a_c is the radius of circular contact region between particles. For the colloidal chain composed of seven PS particles in Fig. 6d-f, κ_0 is found to be approximately 1.1 mN/m. The a_c value can be estimated by the Johnson–Kendall–Roberts (JKR) theory for particle adhesion, given by $a_c = \left(\frac{3\pi R^2 W_{SL}}{2K}\right)^{\frac{1}{3}}$, where $K = \frac{2E_P}{3(1-\nu^2)}$ is the particle elastic modulus and W_{SL} is the adhesion energy between particles. Using the Young's modulus and the Poisson ratio of polystyrene, $E_P = 3.25$ GPa and $\nu = 0.34$, respectively, the particle elastic modulus is estimated to be $K = 2.4$ GPa. At a diluted

[Type here]

electrolyte condition, $W_{SL} = 93.9$ mN/m can be obtained using the Young-Dupré equation $W_{SL} \approx W_{SL}^0 = \gamma_L(1 - \cos \theta_0)$, where the PS-water contact angle¹⁹ is $\theta_0 = 73^\circ$ and the water surface tension is $\gamma_L = 72.7$ mN/m. Using the values of K and W_{SL} , $a_c = 73$ nm and $E_{chain,PS} = 0.05$ GPa are found. For example, the Young's moduli of polystyrene foam and low density polyethylene are ~ 0.005 and ~ 0.2 GPa, respectively.

6) While being a suggestive mechanism, I am unsure that the results of this work would apply to the case of a ligand-receptor interaction (page 18, lines 333-338), as their size (1-10 nanometers) would make the Debye interaction of the kind considered in this paper very weak.

As the reviewer pointed out, it will be challenging to directly measure the interaction force between ligand and receptor due to their small size. However, as mentioned in the conclusion section, current studies on ligand-receptor interactions only consider Coulombic interactions and molecular vdW attractions. Therefore, we suggest that macromolecular-scale D-I attraction resulting from ion rearrangements around the ligand and receptor might also be taken into account, as demonstrated in this study.

Reviewer #2 (Remarks to the Author):

In their study, Lee et al. investigate dipole-induced dipole (D-I) forces between colloids using optical trapping measurements. A permanent dipole is obtained by pinning a polystyrene particle at an oil/water interface. Colloidal aggregates were observed due to the contribution of van der Waals and DI attractive forces. In addition, the DI forces were shown to propagate to the next particles when creating linear chains. The authors have tested many different parameters including the surface functionalization of the polystyrene particles, the nature and concentration of the electrolyte; and further supported their experimental findings with computer simulations. Altogether, I found this fundamental study very interesting both from the quality of the experiments and the clear presentation of the different results. I do believe it is an important contribution that is addressing many opened questions. I recommend its publication for Nature Communications with minor revisions as I believe some points still need to be addressed by the authors. My comments are the following:

We would like to express our appreciation for the positive feedback and the constructive comments of the reviewer. We are pleased to hear that the reviewer found our study on the colloidal D-I force measurements interesting and of high quality. We carefully considered the minor revisions suggested by the reviewer to further improve the quality and clarity of our manuscript.

1. Detection of the oil water interface. The authors use bright field microscopy; would it be possible to directly image the oil water interface using for instance DIC microscopy?

Although it is possible to detect the three-phase contact line position at the bottom coverslip, directly visualizing the oil-water interface at a focal plane would be challenging because the

[Type here]

interface is not perpendicular to the focal plane, given the shape of the sessile drop [Kang et al., *Soft Matter*, 13, 6234 (2017)]. Nonetheless, it is possible to estimate the interface location by moving a trapped particle back and forth near the interface. If the particle is attached to the interface, the optical trap would not detach it from the interface due to the strong attachment energy to the interface [Binks, *Curr. Opin. Colloid Interface Sci.*, 7, 21 (2002)]. In this case, the particle position corresponds to the interface location.

2. Position of the particle at interface. The permanent dipole is created by placing the particle at the oil water interface. Do the authors account for the angle formed by the interface as it will affect their measurements if they only manipulate their particles within the imaging plane as shown in Figure 1d? Ideally, one would try to obtain a contact angle close to 90° and to measure close (but far enough) from the coverslip.

The experiments were conducted at a distance of approximately $40\text{--}50\ \mu\text{m}$ from the coverslip surface, satisfying the “close but far enough” condition), and the sessile drop had a contact angle of $\sim 69^\circ$ with the bottom coverslip. Given that the height of the sessile drop apex from the coverslip is at a millimeter scale, the angle between the focal plane at which P_1^i is located and the interface can be assumed to be approximately the same as $\sim 69^\circ$. Hence, when P_2 approaches P_1^i on the focal plane, the angle between the approach trajectory and the fluid interface may have some impact on the D–I force. However, since the attachment of the two particles occurs even at an angle of $\sim \sin \frac{1}{2} = 30^\circ$ (Fig. S13g), the angle dependence is likely to be at the level of the D–I interaction heterogeneity range depicted in Fig. 2d. Nevertheless, it would be worthwhile to investigate the influence of the approach angle on D–I interaction force in future studies.

Importantly, if the fluid interface were perpendicular to the coverslip, there would be concerns about unquantifiable artifacts resulting from the use of laser tweezers. When the highly focused laser beam from the objective lens traps a particle near the fluid interface, a portion of the Gaussian laser beam may scatter at the interface, leading to potential errors in force measurements. In our current setup, the use of an objective with $\text{NA} = 1.2$ generates a maximum half angle of $\sim 64.5^\circ$ for the con-like laser beam [Park and Furst, *Langmuir*, 24, 13383 (2008)], which is smaller than the contact angle of 69° between the interface and coverslip, thereby eliminating such potential artifacts. This information has been included in the revised manuscript to clarify the experimental setup.

Before: Then, 2 mL of *n*-decane (Acros Organics, USA) was added to cover the sessile drop.

After (p19): Then, 2 mL of *n*-decane (Acros Organics, USA) was added to cover the sessile drop. The contact angle of the sessile drop in *n*-decane was approximately $\sim 69^\circ$.²⁷

Added paragraph (p20): Note that the oil–water interface of the sessile drop formed a contact angle of $\sim 69^\circ$ with the bottom coverslip. The pair interaction measurement was conducted at a distance of approximately $40\text{--}50\ \mu\text{m}$ from the coverslip surface. Given that the height of the sessile drop apex from the coverslip was at a millimeter scale, the angle between the focal plane

[Type here]

at which P_1^i was located and the interface could be assumed to be the same as $\sim 69^\circ$. Although this geometry might affect the D–I interaction due to the non-parallel approach of P_2 to the dipole formed on P_1^i , it could help prevent unquantifiable artifacts resulting from the use of laser tweezers. The highly focused laser beam from the objective lens with $NA = 1.2$ might scatter at the oil–water interface when it trapped a particle near the interface, leading to potential errors in force measurements. However, in the current setup, the objective lens generated a maximum half angle of $\sim 64.5^\circ$ for the con-like laser beam,⁴³ which was smaller than the contact angle of 69° between the interface and coverslip, thereby eliminating such potential artifacts.

Added sentences (p22, SI): Considering the three-phase contact angle $\sim 99.4^\circ$ of the interfacial P_2^i particle and the water-immersed P_1 particle that was almost in contact to the interface, the angle between the interface and the line joining the two particles was approximately $\sin \frac{1}{2} = 30^\circ$. It was notable that the D–I interaction was strong enough to form the dimer when the two particles were not aligned orthogonally with respect to the interface.

Revised Fig. S13g:

Fig. S13. Formation of dimers in large-scale experiments. g, Schematic side view to illustrate the behavior shown in panels a-f.

3. Do the authors have an idea of the contact angle of the particles at the oil/water interface from the literature? It is certainly an important parameter as it defined the extent of the protrusion of the particle in the oil-phase. How does it compare to their simulations?

The contact angle is a critical parameter that affects the interparticle interactions and dipole strength, as mentioned by the reviewer. The three-phase contact angle of the SPS particle at the oil–water interface is approximately 99.4° [Choi et al., ACS Appl. Polym. Mater. 2, 1304, 2020], while the simulation result is around 70.8° . This difference of $\sim 28.6^\circ$ might arise from the differences in surface roughness and surface charge distribution between the experimental particles and the simulated nanoparticles. However, we do not anticipate that this difference in contact angle will significantly affect the investigation of elucidating ion rearrangement and induced dipole formation in the diffuse layer through CGMD simulations. We revised the manuscript and SI accordingly and added a section to the revised SI to determine the three-phase contact angle in simulations.

Added Fig. S8:

[Type here]

Fig. S8. Estimation of the three-phase contact angle θ of a simulated SPS nanoparticle at the oil–water interface. **a**, Molecular configurations of the SPS particle at the interface with the particle diameter d and contact line diameter d_c , depicted in blue and red lines, respectively. Water molecules were not shown for clarity. **b**, Number density of the CG SPS beads in the water phase as a function of the radial distance from the particle center. **c**, Number density of the CG SPS beads of the cross-sectional area by the oil–water interface as a function of the radial distance from the cross-section center.

Added sentence (p6): The three-phase contact angle of the SPS particle at the oil–water interface was approximately 99.4° .²³

Before: Based on these simulation conditions, when a PS particle is at the oil–water interface, the dipole moment of P_1^i perpendicular to the interface is $\mu_{1,MD} = 7.12 \times 10^{-9} \text{ pC} \cdot \mu\text{m}$.

After (p12): Based on these simulation conditions, when the PS particle was at the oil–water interface, the three-phase contact angle was $\sim 70.8^\circ$ (Fig. S8) and the dipole moment of P_1^i perpendicular to the interface was $\mu_{1,MD} = 7.12 \times 10^{-9} \text{ pC} \cdot \mu\text{m}$.

Added paragraph (p17, SI): The three-phase contact angle θ of a simulated nanoparticle at the oil–water interface was calculated using $\sin \theta = d_c/d$, where d and d_c are the particle diameter and the diameter of the cross-sectional area of the particle at the interface, respectively (Fig. S8a). To estimate its effective diameter, the number densities of the coarse-grained (CG) SPS beads were analyzed along the radial direction from the particle center for d (Fig. S8b) and from the cross-section center for d_c (Fig. S8c). The effective diameter was determined at the second inflection point of the number density. It was found that $d_c \approx 20.4 \text{ nm}$ and $d \approx 21.6 \text{ nm}$, resulting in $\theta \approx 70.8^\circ$.

4. Dipole-dipole interactions. If the dipole-dipole forces are the product of the asymmetric surface charge dissociation, I would expect them to strongly depend on the ionic strength. Did the authors perform such measurements at different ionic strength or for different electrolytes (i.e. different Debye lengths)?

[Type here]

As the reviewer noted, the sensitivity of the dipole-dipole interactions to the ionic strength of the solution has been well documented in previous studies [Park et al., *Langmuir* 24, 1686 (2008); Reyanert et al., *Langmuir* 22, 4936 (2006); Frydel et al., *Phys. Rev. Lett.* 99, 118302 (2007)]. The pair interaction forces at the oil-water interface were quantitatively measured using optical laser tweezers over a range of electrolyte concentrations [Park and Furst, *Soft Matter* 7, 7676 (2011)]. In our study, we used a 10 mM NaCl solution, which was chosen as it was favorable for the adsorption of P_1 particles to the interface by reducing the electrostatic repulsion between the negatively charged fluid interface and the particle [Kang et al., *Soft Matter* 13, 6234 (2017); Kang et al., *Langmuir* 34, 8839 (2018)]. However, this solution was not favorable for dimer formation in the 10 mM NaCl water-only condition (Fig. 3c). Therefore, we focused on measuring the D–I interaction forces under the optimal condition of 10 mM NaCl, where we could confirm the presence of D–I attraction and separate its contribution from vdW attraction.

5. Figure 2d: force measurement. The different contributions to the measured forces are not clearly visible from the linear scale. I would suggest the authors to use a logarithmic scale for the x-axis. In addition, they should detail the different contributions to the total force. What is the expression they used for F_{el} and for F_{vdw} from the DLVO theory? I would suggest the authors to add a paragraph in the supporting with the different parameters and assumptions that were made (surface potential=zeta potential? Effective Hamacker constant, radius, Debye length). An additional force measurement in water at lower ionic strength should be provided to demonstrate the robustness of the force measurements.

We measured the D–I interaction forces for 72 pairs and included a revised version of Fig. 2d, with the x-axis converted to a logarithmic scale. We also added equations and parameter values for the DLVO interactions (F_{el} and F_{vdw}) used to obtain the results in Fig. 2d and 2e in the revised SI.

To demonstrate the robustness of our force measurements, we added a plot of the forces measured for nine different pairs in the 10 mM water-only condition in Fig. S3.

Before: We measured the forces for ~30 pairs due to the interaction heterogeneity present when measuring the colloidal interaction forces.^{22, 24, 28} Among these pairs, the force profiles (F_{D-I}) of seven with considerable attraction magnitude are shown in Fig. 2d. The other pairs did not show measurable attractive forces. The measured force was fitted with $F_{fit} = F_0 \left(\frac{d}{r}\right)^b$. Two-parameter fitting (inset in Fig. 2d) resulted in average values of $\langle F_0 \rangle = -0.208 \pm 0.133$ pN and $\langle b \rangle = 6.913 \pm 0.854$ with $\langle \chi^2 \rangle = 1.14 \times 10^{-3}$, and one-parameter fitting with a fixed power law exponent of $b = 7$ resulted in values of $\langle F_0 \rangle = -0.168 \pm 0.097$ pN with $\langle \chi^2 \rangle = 1.19 \times 10^{-3}$.

After (p7): We measured the forces for 290 pairs due to the interaction heterogeneity present when measuring the colloidal interaction forces.^{22, 23, 28} Among these pairs, the 72 force profiles (F_{D-I}) with considerable attraction magnitude are shown in Fig. 2d. The other pairs did not show measurable attractive forces. The measured force was fitted with $F_{fit} = F_0 \left(\frac{d}{r}\right)^b$. Two-

[Type here]

parameter fitting (insets in Fig. 2d) resulted in average values of $\langle F_0 \rangle = -0.44 \pm 0.17$ pN and $\langle b \rangle = 7.18 \pm 1.22$ with $\langle \chi^2 \rangle = 9.87 \times 10^{-3}$.

Revised Fig. 2d,e:

Fig. 2. Direct measurements of the D–I interaction force. d, Measured D–I interaction force profile (F_{D-I}) and comparison with the DLVO forces (F_{el} and F_{vdW}). The water phase contains 10 mM NaCl. The x-axis of the graph is on a logarithmic scale. The red solid line represents a fitted curve that uses the mean values of the two fitting parameters, $\langle F_0 \rangle = -0.44$ and $\langle b \rangle = 7.18$. The black dotted lines indicate the guideline for $F \sim r^{-7}$. The insets show histograms of the values of the two fitting parameters for 72 pairs. The force profiles display before the paired particles come into contact. **e**, Magnified force profiles in the short-range separation, and the x-axis is on a logarithmic scale.

Revised Fig. S3:

[Type here]

Fig. S3. Pair interaction forces for nine different pairs suspended in 10 mM NaCl water. The grey circles represent the force profile for each pair, with the z-axis error bars indicating thermal fluctuations observed while holding the particles with optical tweezers. The red circles represent the average force profile over the nine pairs, with the error bars indicating the corresponding standard deviation. The x-axis is shown in log scale. The inset provides a magnified view of the force profile near separations.

Before (SI): When two SPS particles were dispersed in 10 mM water, a nonsignificant negative force was detected near separations (Fig. S2). Considering the errors caused by image analysis and thermal fluctuation of the particles, it seems unreasonable to conclude that the negative force can be attributed to the secondary energy minimum resulting from the classical DLVO theory. In addition, after the two particles approached each other closely, optical laser tweezers could readily separate them without the presence of measurable forces. The result of this force measurement in 10 mM NaCl water was consistent with the dimer formation probability in the same fluid condition, in which only six of 256 pairs formed aggregate dimers ($P_f \approx 2\%$), as in Fig. 3c.

After (p11, SI): Pair interaction forces were measured for SPS particles in 10 mM NaCl water to compare them with the D–I forces. To measure the interaction force between two particles in the 10 mM NaCl water-only condition, one particle P_1 was fixed with a stationary trap, and the other particle P_2 approached the P_1 particle with a translational trap stepwise. The displacements Δx of the stationary particle from its equilibrium position were measured as a function of the particle separation r , and Δx was converted to the pair interaction force using $F(r) = \kappa_t \Delta x(r)$.¹⁴ A total nine different pairs were measured, and a nonsignificant negative force was detected near separations, as shown in Fig. S3. Note that the positive force in Fig. S3 represents that the optical traps holding the two particles could not push more closely together, and the P_1 particle was pushed backward in the approach direction of the P_2 . In addition, after the two particles approached each other closely, optical laser tweezers could readily separate them without the presence of measurable forces. The result of this force measurement in 10 mM NaCl water was consistent with the dimer formation probability in the

[Type here]

same fluid condition, in which only six of 256 pairs formed aggregate dimers ($P_f \approx 2\%$), as in Fig. 3c.

Added paragraph (p12, SI): The Derjauin-Landau-Verwey-Overbeek (DLVO) interaction theory provides a framework for describing colloidal interactions in an aqueous phase.^{12, 20} For two spherical colloids with an equal diameter d , the vdW interaction can be expressed as $U_{vdW} = -\frac{A_H d}{24(r-d)}$, where $A_H = 1.4 \times 10^{-20} J$ is the Hamaker constant between two PS particles interacting across water.¹² The double layer interaction between two colloidal spheres is given by $U_{el} = 32 \times 10^3 \pi d k_B T I N_A \kappa^{-2} \gamma_0^2 \exp(-\kappa(r-d))$ for $\kappa\left(\frac{d}{2}\right) > 10$, where $I = 10$ mM is the ionic strength, N_A is Avogadro's number, $\gamma_0 = \tanh\frac{e\psi}{4k_B T}$ is the Gouy-Chapman parameter, $\psi = -57.5$ mV is the particle zeta-potential, e is the elementary charge, and $\kappa = \sqrt{\frac{2000e^2 I N_A}{\epsilon_W \epsilon_0 k_B T}}$ is the inverse Debye screening length.^{20, 21} Note that the U_{el} equation agrees well with the numerical solution of the non-linear Poisson-Boltzmann equation, even at small interparticle separations ($\kappa(r-d) < 1$).²² The corresponding force can be calculated numerically as the derivative of the potential energy with respect to interparticle separation, i.e., $F = -\frac{dU}{dr}$. The DLVO interaction forces between two particles, where one particle is attached to the oil-water interface and the other is immersed in water (Fig. 2d and 2e), were also estimated using the above equations.

6. The colloidal D-I forces seem strongly related to the asymmetric distribution of counterions and to be very much dependent on the particle charge, size and electrolyte concentrations. How much the comparison with the molecular polarizability which is only described in the paper as a differences of permittivity makes sense?

We acknowledge that the colloidal D-I force is influenced by various variables, including the dipole strength of interface-attached P_1^i particle, particle charge, size, and ionic strength, as pointed out by the reviewer. Please note that the experimentally measured dipole strength value μ_1 of the P_1^i particle and the measured D-I force were obtained under the influence of these variable conditions (please refer to p5-8 in SI for a detailed explanation). To analyze the results, we fitted the measured D-I force using the equation $F_{D-I} = F_0 \left(\frac{d}{r}\right)^b$ and obtained the force magnitude $F_0 = -\frac{3\mu_1^2 \alpha_2'}{2\pi \epsilon_0 \epsilon_w d^7}$. We used this F_0 value to predict the polarizability volume of P_2 particle and compare it with the particle volume. This relationship is similar to the one theoretically proposed in molecular Debye interactions. We added the detailed description for all equations used in the work to clarify this point.

Before: The D-I interaction force was given by $F_{D-I} = -\frac{dU_{D-I}}{dr} = F_0 \left(\frac{d}{r}\right)^7$, where $F_0 = -\frac{3\mu_1^2 \alpha_2'}{\pi \epsilon_0 \epsilon_w d^7}$. The permanent dipole moment μ_1 for the interface-trapped particles can be

[Type here]

obtained using the self-potential method²⁴ by measuring the dipole-dipole pair interaction forces between i and j particles at a planar oil–water interface $F_{D-D} = \frac{3\mu_i\mu_j}{8\pi\epsilon_0\epsilon_{oil}d^4} \left(\frac{d}{r}\right)^4$ (Fig. S1a). The mean value for 18 particles was $\langle\mu_1\rangle \times 10^4 = 4.5 \pm 2.0$ pC· μm at the same fluid condition (Fig. S1b and S1c). Consequently, the polarizability volume was $\alpha'_2 = \alpha'_{2,exp} \approx 1.49$ μm^3 at $F_0 = -0.208$ pN, which was in order-of-magnitude agreement with the theoretical prediction of the molecular polarizability volume in vacuum, $\alpha'_{2,theory} \sim \left(\frac{d}{2}\right)^3 \approx 3.2$ μm^3 . Notably, a dielectric sphere in a solvent medium with dielectric constants ϵ_p and ϵ_w can be polarized by an electric field, and the effective polarizability volume α'_{eff} can be reduced by a factor of $\left|\frac{\epsilon_p - \epsilon_w}{\epsilon_p + 2\epsilon_w}\right|$,⁷ resulting in $\alpha'_{eff} \approx 1.54$ μm^3 , which shows excellent agreement with the experimental value $\alpha'_{2,exp}$.

After (p7): The D–I interaction force can be expressed as $F_{D-I} = -\frac{dU_{D-I}}{dr} = F_0 \left(\frac{d}{r}\right)^7$, where $F_0 = -\frac{3\mu_1^2\alpha'_2}{2\pi\epsilon_0\epsilon_w d^7}$ and ϵ_w represents the water dielectric constant. The permanent dipole moment μ_1 for the interface-trapped particles can be obtained using the self-potential method²³ by measuring the dipole–dipole pair interaction forces between i and j particles at a planar oil–water interface $F_{D-D} = \frac{3\mu_i\mu_j}{8\pi\epsilon_0\epsilon_{oil}d^4} \left(\frac{d}{r}\right)^4$, where ϵ_{oil} is the decane dielectric constant (Fig. S2a).^{20, 29} The mean value for 18 particles was found to be $\langle\mu_1\rangle \times 10^4 = 4.5 \pm 2.0$ pC· μm at the same fluid condition (Fig. S2b and S2c). The polarizability volume was then calculated as $\alpha'_2 = -\frac{2\pi\epsilon_0\epsilon_w d^7 F_0}{3\mu_1^2} = \alpha'_{2,exp} \approx 6.29$ μm^3 at $F_0 = -0.44$ pN, which was in order-of-magnitude agreement with the theoretical prediction of the molecular polarizability volume in vacuum, $\alpha'_{2,theory} \sim \frac{4\pi}{3} \left(\frac{d}{2}\right)^3 \approx 13.6$ μm^3 . Additionally, a dielectric sphere in a solvent medium with dielectric constants ϵ_p and ϵ_w can be polarized by an electric field, and the effective polarizability volume α'_{eff} can be reduced by a factor of $\left|\frac{\epsilon_p - \epsilon_w}{\epsilon_p + 2\epsilon_w}\right|$,⁷ resulting in $\alpha'_{eff} \approx 6.47$ μm^3 , which shows excellent agreement with the experimental value $\alpha'_{2,exp}$.

Added paragraph (p5, SI):

General formulas for interactions between molecules.^{11, 12}

When two permanent point dipoles μ_1 and μ_2 are fixed at a mutual orientation angle of θ and separated by a distance r in a medium, their dipole-dipole (D–D) potential energy can be expressed as $U_{D-D, fixed} = \frac{\mu_1\mu_2 f(\theta)}{4\pi\epsilon_0\epsilon_r} r^{-3}$, where ϵ_0 is the vacuum permittivity, ϵ_r is the dielectric constant of the medium, and $f(\theta) = 1 - 3\cos^2\theta$. The r^{-3} dependence arises from the field strength of the two point dipoles (r^{-1} dependence) and the magnitude of each dipole decreasing as r increases (r^{-2} dependence). For two freely rotatable dipoles, their relative orientation is constrained by the interaction strength (r^{-3} dependence). The Keesom interaction, which is the first contribution to the van der Waals (vdW) interaction, is always attractive and can be described by combining the Boltzmann equation (r^{-3} dependence) with $U_{D-D, fixed}$: $U_{D-D, free} = -\frac{2}{3k_B T} \frac{\mu_1^2\mu_2^2}{(4\pi\epsilon_0\epsilon_r)^2} r^{-6}$, where k_B is the Boltzmann constant and T is the temperature.

[Type here]

When nonpolar molecules are exposed to an electric field, their electronic distribution and nuclear positions are distorted, inducing a temporary dipole moment. For moderate field strengths E , the magnitude of the induced dipole moment μ^* is linearly proportional to E and can be described by $\mu^* = \alpha E$, where α represents the molecular polarizability. The polarizability volume α' can be expressed as $\alpha' = \frac{\alpha}{4\pi\epsilon_0\epsilon_r} = \frac{\mu^*}{4\pi\epsilon_0\epsilon_r E}$. A permanent dipole μ_1 can induce a dipole moment μ_2^* in a nonpolar, polarizable molecule. The interaction between the dipole and the induced dipole (D–I) is attractive and is referred to as the Debye interaction, which is the second contribution to the vdW interaction. The Debye interaction potential is given by $U_{D-I} = -\frac{1}{2}\alpha_2 E^2$, where $E = \frac{\mu_1(1+3\cos^2\theta)^{\frac{1}{2}}}{4\pi\epsilon_0\epsilon_r r^3}$ is the electric field generated by μ_1 . When the Debye interaction is not strong enough to mutually orient the molecules, the angle average of $\cos^2\theta = \frac{1}{3}$ is used to describe U_{D-I} , which is given by $U_{D-I} = -\frac{\mu_1^2\alpha_2}{(4\pi\epsilon_0\epsilon_r)^2} r^{-6} = -\frac{\mu_1^2\alpha_2'}{4\pi\epsilon_0\epsilon_r} r^{-6}$. The inverse sixth power arises from the r^{-3} dependence of the magnitude of the induced dipole, which is weighted by the r^{-3} dependence of the interaction between the dipole and the induced dipole. The corresponding D–I interaction force is given by $F_{D-I} = -\frac{dU_{D-I}}{dr} = F_0 \left(\frac{d}{r}\right)^7$, where $F_0 = -\frac{3\mu_1^2\alpha_2'}{2\pi\epsilon_0\epsilon_r d^7}$. When the two molecules are mutually oriented with $\theta = 0^\circ$ and thus $\cos^2\theta = 1$, the attraction is doubled, i.e., $U_{D-I}(\theta = 0^\circ) = -\frac{\mu_1^2\alpha_2'}{2\pi\epsilon_0\epsilon_r} r^{-6}$.

7. D-I forces. The D-I forces together with vdW forces seem to hold the assembly together. In addition, at lower ionic strength I would still expect repulsive screened Coulomb interactions to take place. Is it the reason why Pf decreases at lower ionic strength? Figure 3c may indicate that the particles hold together due to vdW forces at high ionic strength. Was it possible to measure Pf at higher ionic strength to show that ultimately Pf at interface becomes equal to Pf in water?

The low probability of dimer formation in water-only environments at low ionic strength is due to the dominance of double layer electrostatic repulsion (F_{el}) over vdW attraction. This means that the energy barrier created by the DLVO interaction between two particles is difficult to overcome at such low concentrations. Although the theoretically calculated DLVO force in Fig. 2e does not show an energy barrier, the discrepancy between the theoretical calculations and the experimental results may be due to particle surface roughness and other factors [Suresh and Walz, J. Colloid Interface Sci. 183, 199 (1996); Suresh and Walz, J. Colloid Interface Sci. 196, 177 (1997); Pantina and Furst, Langmuir 20, 3940 (2004)].

In the decane/water environment with 10 mM NaCl, ion rearrangement around the P_1^i and P_2 particles induces colloidal D–I interaction force and likely reduces F_{el} , allowing the particles to approach spontaneously without the need to overcome the energy barrier at a relatively long distance. When the two particles are in close proximity, the D–I interaction induced by the ion rearrangement decreases, and dimer formation is likely driven by the colloidal vdW.

[Type here]

The dimer formation probability decreases in the decane/water environment with 1 mM NaCl, as there may not be a sufficient number of ions present to generate a strong induced dipole on the P_2 particles. This decreased chance of inducing a dipole can result in F_{el} preventing dimer formation.

As the reviewer mentioned, the dimer formation probability is expected to increase in high salt concentration decane/water environments, similar to water-only environments (Fig. 3c). However, determining the relative contributions of the presence of D–I interaction and the reduction of F_{el} to dimer formation in such higher salt conditions would be challenging, making it difficult to reveal the presence of D–I interaction in decane/water conditions. Therefore, we focused on performing intensive experiments at the 10 mM NaCl condition, where F_{el} at the 10 mM NaCl water-only condition is strong enough and the dimer formation probability is sufficiently low (Fig. 3c). Additionally, as the reviewer suggested, the dimer formation probability is expected to be close to 1 in both water-only and decane/water environments with high salt concentrations. However, such high concentration experiments pose a challenge because the particles become strongly attached to the surface of the bottom glass substrate, making it difficult to detach them using optical laser tweezers to conduct further experiments. We clarified this point in the revised manuscript.

Before: The reduction in P_f with a decrease in C_{NaCl} (Fig. 3c) revealed the important role of the surrounding ions in forming the induced dipole on the water immersed particle. Conversely, when the SPS particles were completely immersed in water, P_f was 0 (0/10) at $C_{NaCl} = 1$ mM and 0.02 (6/256) at $C_{NaCl} = 10$ mM (Fig. 3c). This is consistent with the measured force profile without showing a considerable well depth (Fig. S2).

After (p10): The reduction in P_f with a decrease in C_{NaCl} (Fig. 3c) indicates the important role of the surrounding ions in forming the induced dipole on the water immersed particle. Conversely, when the SPS particles were completely immersed in water and brought as close as possible using optical laser tweezers, P_f was 0 (0/10) at $C_{NaCl} = 1$ mM and 0.02 (6/256) at $C_{NaCl} = 10$ mM (Fig. 3c), which is consistent with the measured force profile without showing a considerable well depth (Fig. S3). P_f increased significantly in the water-only environment with higher NaCl concentrations (Fig. 3c), as F_{el} was sufficiently screened, leading to a relatively greater contribution of the vdW force to the dimer formation. Although it was expected that P_f would also increase in such high NaCl decane/water environments, determining the relative contributions of the D–I interaction and the reduced F_{el} to dimer formation in such conditions would be challenging, and it would not be appropriate to claim the presence of D–I interaction.

8. Following the classical DLVO theory, it could maybe have been interesting to measure the stability of different particles to extract the surface potential from the critical coagulation concentration.

We thank the reviewer for the suggestion. While we agree that investigating colloidal stability by extracting surface potential from critical coagulation concentrations could be interesting and

[Type here]

applicable to our decane/water environment, our study was primarily focused on demonstrating the presence of D–I interaction at colloidal scales. We achieved this by performing a variety of experiments and simulations, including pair interaction measurement, dimer formation probability measurement, colloidal chain formation experiment, CGMD simulation, and large interface scale experiment.

9. The measurements performed with different salt and particles in Figure 3a are very intriguing. Could they further comment on the fact that 10 mM LiCl and 10 mM CaCl₂ provide similar results?

We aimed to investigate the impact of ion mobility on the strength of D–I interaction and the probability of dimer formation over time using LiCl and CaCl₂ electrolytes. Generally, Li⁺ ions have higher mobility and can move more quickly in solution than Ca²⁺ ions. While our preliminary experimental results did not provide sufficient evidence regarding the effect of different electrolyte use on the dimer formation, further careful studies using a variety of ions and statistically significant experiments would be valuable for exploring the effect of ion type on the D–I interaction. Therefore, we added this point in the revised manuscript.

Before: Based on these results, we assumed that various factors, such as surface functional groups, heterogeneous surface charge distribution, surface roughness, and ion mobility, could affect the D–I interactions and dimer formation probability. Therefore, quantitative in-depth investigations on each factor should be conducted in subsequent studies.

After (p10): Based on these results, it is assumed that various factors such as surface functional groups, surface roughness, and ion types and mobility could affect the D–I interaction and dimer formation probability. Therefore, quantitative in-depth investigations on each factor should be conducted in subsequent studies.

Before: Further investigation of the D–I interactions will be performed on evaluating colloidal chain-chain interactions and the effects of electrolytes and surfactants on chain micromechanics.

After (p19): Further investigation will be performed on evaluating the effects of various electrolytes and surfactants on D–I interactions and the micromechanics of colloidal chains.

10. Why are the particles in the simulations so rough? In practice, the PS particles should be only surface functionalized and I would expect them to be rather smooth, particularly when their molecular roughness is compared to their diameter.

We thank the reviewer for the valuable comment. As the reviewer pointed out, it is indeed possible to produce a large spherical particle and functionalize its surface. However, this approach could lead to an unclear charge state of the spherical particle. Therefore, we chose to

[Type here]

utilize polymer chains to create the SPS particle in our study. Specifically, the SPS particle was composed of 32 chains of SPS polymer, each with a 500-mer (equivalent to 99,528 Da). These chains spontaneously formed a particle to lower the surface energy, resulting in the SPS particle with a diameter of ~22 nm. This particle size was only five times larger than the radius of gyration of the SPS polymer, which was about 4.4 nm. Therefore, the surface roughness of our simulation model was due to the small size of the SPS particle compared to the polymer chain length. However, this condition was sufficient to evaluate the D–I interaction phenomena.

11. Colloidal chain. Please provide the Euler-Bernoulli equation in the supporting information.

In the revised SI, we included the Euler-Bernoulli equation and further analyses of colloidal chains during bending.

Added paragraph (p9, SI):

Bending experiment of colloidal chains.^{17, 18}

The deflection y of a colloidal chain with a length L under an applied load F can be described by the Euler-Bernoulli beam equation, $y(x) = \frac{F}{6EI}(3Lx^2 - x^3)$, where E is the Young's modulus and I is the area moment of inertia. The bending rigidity of a single-bonded colloidal linear chain is expressed as $\kappa_{chain} = \frac{F_{bend}}{\delta}$. In Fig. 6f, the linear regression of the force profile resulted in $\kappa_{chain} \approx 0.41$ pN/ μ m. The single-bond rigidity κ_0 can be defined as $\kappa_0 = \kappa_{chain} \left(\frac{s}{R}\right)^3 = \frac{3\pi a_c^4 E_{chain}}{4R^3}$, where s is the chain contour length, R is the particle radius, E_{chain} is the Young's modulus of colloidal chain, and a_c is the radius of circular contact region between particles. For the colloidal chain composed of seven PS particles in Fig. 6d-f, κ_0 is found to be approximately 1.1 mN/m. The a_c value can be estimated by the Johnson–Kendall–Roberts (JKR) theory for particle adhesion, given by $a_c = \left(\frac{3\pi R^2 W_{SL}}{2K}\right)^{\frac{1}{3}}$, where $K = \frac{2E_p}{3(1-\nu^2)}$ is the particle elastic modulus and W_{SL} is the adhesion energy between particles. Using the Young's modulus and the Poisson ratio of polystyrene, $E_p = 3.25$ GPa and $\nu = 0.34$, respectively, the particle elastic modulus is estimated to be $K = 2.4$ GPa. At a diluted electrolyte condition, $W_{SL} = 93.9$ mN/m can be obtained using the Young-Dupré equation $W_{SL} \approx W_{SL}^0 = \gamma_L(1 - \cos \theta_0)$, where the PS-water contact angle¹⁹ is $\theta_0 = 73^\circ$ and the water surface tension is $\gamma_L = 72.7$ mN/m. Using the values of K and W_{SL} , $a_c = 73$ nm and $E_{chain,PS} = 0.05$ GPa are found. For example, the Young's moduli of polystyrene foam and low density polyethylene are ~0.005 and ~0.2 GPa, respectively.

As the chain bends, aren't the DI-forces expected to decrease as the permanent dipole maintain is orientation? Could the authors please comment on this?

Our experimental and CGMD simulation results indicate that the D–I interaction force is generated by the electric field produced by the dipole of the interface-attached P_1^i particle, which rearranges ions around the water-dispersed P_2 particle, leading to the formation of an

[Type here]

induced dipole. This ion rearrangement is expected to reduce the contribution of double layer repulsion in the DLVO interaction and facilitate spontaneous approach between the two particles at relatively long distances. As the two particles approach nearly in contact, it is anticipated that the ions between them become sparser, leading to a decrease in the D–I force contribution, and the formation of colloidal chains in this range is primarily driven by the colloidal vdW. Therefore, we believe that considering the change in the D–I force contribution is not crucial once the colloidal chain has been formed. We clarified this point in the revised manuscript.

Before: We speculated that the ion distribution between two particles can be sparse when they are sufficiently close. As a result, the diffuse layer polarization around P_2 is reduced, and the two particles can attach to each other under the influence of vdW.

After (p12): The simulations may also provide insight into whether the D–I interaction persists as the particles get closer or whether it disappears at a certain distance due to sparse ion distribution between them. If the two particles are nearly in contact, the diffuse layer polarization around P_2 may be reduced, and they may primarily attach to each other under the influence of vdW.

Before: The simulation results showed overall consistency with the experimental observations and the theoretical predictions from the DLVO interactions, as discussed in Fig. 2d,e. The D–I interaction initiated attraction in relatively long-range separations, and vdW attraction became dominant in relatively short-range separations. When two SPS particles were completely immersed in water, they did not show such an asymmetric anion distribution (Fig. S7). Consequently, attraction and dimer formation were not observed in the water condition, demonstrating the crucial role of the D–I interaction in initiating the attraction over long-range separations at the oil–water interface.

After (P13): The simulation results demonstrated that the D–I interaction played a critical role in initiating attraction in long-range separations, whereas vdW attraction became dominant in short-range separations, consistent with the experimental observations and the theoretical predictions from the DLVO interactions discussed in Fig. 2d,e. To further investigate the importance of the D–I interaction, a CGMD simulation was conducted for two water-immersed particles using a similar initial condition to that used in Fig. 4a, where the particles were initially separated with a finite surface-to-surface distance. The results showed that the asymmetric anion distribution was not observed (Fig. S9), and thus, attraction and dimer formation did not occur in the water-only condition, highlighting the crucial role of the D–I interaction in initiating attraction over long-range separations at the oil–water environment. In contrast, when the two particles were initially brought as close as possible, the preformed dimer remained stable (Fig. S10), indicating that the initial state of the simulation had already fallen into the secondary energy minimum.

The authors need to define κ_t and δd more precisely in their manuscript. Does F_{bend} can be compared to the total force between two particles?

[Type here]

As depicted in Fig. 6e, the bending force F_{bend} acting on the colloidal chain was proportional to the displacement Δd from the P_7 's equilibrium position when the P_1^i and P_7 particles were fixed using optical traps, and the P_4 particle was moved either upwards or downwards by a distance of δ . An explanation of the formula $F_{bend}(\delta) = \kappa_t \Delta d$ used in Fig. 6f has been added to the revised manuscript.

The formation of the colloidal chain was initiated by the D–I interaction when the particles were relatively far apart, while vdW became prominent when the particles were in proximity. Accordingly, the bending properties of the chain observed in our work were consistent with the typical behaviors of colloidal chains formed by vdW at high electrolyte concentrations, as reported in previous studies [Pantina and Furst, Phys. Rev. Lett. 94, 13801 (2005); Pantina and Furst, Langmuir 20, 3940, 2004]. Therefore, we believe that the chain bending force is not closely related to the D–I force.

Before: In addition, the corresponding force profiles during the bending and relaxing events were similar (Fig. 6f), indicating no small-scale rearrangements between the particles due to a critical bending moment.

After (p17): In addition, the corresponding force could be obtained by using $F_{bend}(\delta) = \kappa_t \Delta d$, where δ is the chain deflection and Δd is the displacement of the P_7 particle from its equilibrium position. The resulting profiles during the bending and relaxing events were similar (Fig. 6f), indicating no small-scale rearrangements between the particles due to a critical bending moment.

What do they learn from the bending force at rupture?

The bending experiment in Fig. 6d-f demonstrated that the formed colloidal chain could withstand the bending force without experiencing rolling or sliding, which is similar to the behavior of singly bonded colloidal aggregates of PMMA particles formed by the vdW force, as previously reported [Pantina and Furst, Phys. Rev. Lett. 94, 13801 (2005)]. In some cases, small-scale rearrangements followed by rupture occur in colloidal chains, indicating that the particles are in a shallow secondary energy minimum that can be detached by optical tweezers [Pantina and Furst, Langmuir 20, 3940 (2004)]. Therefore, this consistency confirms our claim in the paper that the D–I force dominates at relatively long distances, while the vdW force dominates when the particles are nearly in contact. We clarified this point in the revised manuscript.

Before: Local rearrangements between the particles and chain rupture were observed when a critical bending moment was applied to a colloidal chain in another bending experiment (pentamer in Fig. S9).

After (p18): Local rearrangements between the particles and chain rupture were observed when a critical bending moment was applied to a colloidal chain in another bending experiment (pentamer in Fig. S12). These results indicated that the particles was in a shallow secondary

[Type here]

energy minimum formed by the colloidal vdW force that could be detached by optical tweezers.³²

12. Line 296. “in 10 mM water” should read “in 10 mM NaCl water”

We have revised this part in the revised manuscript.

Before: The same experiment was performed but all particles were in 10 mM water, and no particle attachment occurred (Fig. S8).

After (p17): The same experiment was performed but all particles were in 10 mM NaCl water, and no particle attachment occurred (Fig. S11).

Reviewer #3 (Remarks to the Author):

This manuscript describes measurements and simulations of the interaction between a charged colloidal sphere embedded in an oil-water interface and one or more similar spheres dispersed in the aqueous phase. This system is expected to display Debye attractions between the permanent electric dipole moment of the interfacial particle and induced dipole moments in the neighboring spheres. As the authors point out, colloidal Debye interactions have not been reported previously, nor have their interesting collective effects been described. In addition to reporting optical-tweezer measurements of interparticle forces, the authors use optical tweezers to assemble particles into flexible chains that are anchored by an interfacial particle. The experimental study is supported by molecular dynamics simulations that demonstrate Debye attractions between interfacial and bulk particles under conditions where van der Waals attractions are weak. The subject of this contribution meets the standards of novelty and importance for publication in Nature Communications.

The current manuscript, however, does not provide nearly enough information for the reader to assess whether or not its conclusions are valid. Without this information, I would not recommend publication in any journal. Assuming it can be provided and supports the current manuscript's interpretation, I would be inclined to recommend publication in Nature Communications.

We thank the reviewer for the positive feedback on the novelty and importance of our work. We provide the necessary additional information to ensure the validity of our conclusions.

1. The paper should provide details on the instrument. Is it based on a commercial microscope? If so, which one? If the instrument was custom-built, the text should provide information on its components and its layout. What is the magnification of the imaging system (in micrometers per pixel)? What is the exposure time of the camera? What is the power and wavelength of the trapping laser?

[Type here]

2. How were particle positions measured? Was a standard software package used, or custom software? What algorithm was used to localize the particles' centroids?

For the comments 1 and 2, we included all the requested information in the revised SI.

Added paragraph (p2, SI):

Optical laser tweezer setup.

To set up the optical laser tweezers, we used an inverted microscope (Ti-U, Nikon, Japan) along with a 10W CW Nd:YAG laser with a 1064 nm wavelength.¹⁻³ The laser beam passed through an acousto-optic deflector (AOD, Opto-electric DTSXY-400-1064 2D, AA Opto Electronic, USA) before entering a water immersion objective (CFI Plan Apochromat VC 60 \times , Nikon, Japan) with a numerical aperture of NA = 1.2 and a working distance of \sim 300 μ m. By focusing the beam on a specific focal plane, an optical trap was generated. The x and y positions of the trap were adjusted by diffracting the laser beam through the AOD, which was operated with LabVIEW software. Multiple optical traps were generated by time-sharing a single beam using the AOD, and their trap stiffnesses were equalized by implementing a consistent pause time at each discrete trap position.³ While each trap stiffness increased with laser power, it decreased with the number of time-shared traps generated. To measure the pair interaction force, we typically generated a total of 10 optical traps, each receiving an allocated laser power of \sim 7 mW, making the total measured laser power \sim 70 mW. The laser power was directly measured using an optical power meter (PM100D, Thorlabs, USA) positioned above the objective.

Added paragraph (p4, SI):

Image analysis.

A microscopic movie was recorded using a charge-coupled device (CCD) camera (Hitachi, KP-M1AN, Japan) and/or a complementary metal-oxide-semiconductor (CMOS) camera (CS505CU, Thorlabs, USA), installed in the Ti-U inverted microscope, at a rate of 30 frames per second (fps). The exposure time and magnification were 16.7 ms and 4.41 pixel/ μ m for the CCD camera and 29.998 ms and 17.55 pixel/ μ m for the CMOS camera, respectively. The recorded movie was saved as a sequence of microscopic images using the ImageJ software.⁸ Particle positions were analyzed using a standard particle tracking routine⁹ implemented with MATLAB.¹⁰ Notably, for the pair interaction force measurements reported in Fig. 2d, cases where two particles did not contact each other were only considered. As a result, the possibility of any overlapping effects of the two particles causing artifacts during the image analysis process could be minimized.

Does that algorithm account for the overlapping diffraction patterns of particles near contact? This is a particularly important point because such overlaps are known to introduce artifacts into inferred interaction potentials. A standard reference on this point is

Baumgartl J, Bechinger C. On the limits of digital video microscopy. *EPL (Europhysics Letters)* 71, 487 (2005).

In explaining the measurement technique, the authors also should specify the precision and accuracy with which their algorithm tracks their particles' three-dimensional positions.

[Type here]

Projection errors due to out-of-plane motions are critically important for colloidal force measurements near contact.

According to Baumgartl and Bechinger, the image analysis error due to optical distortion, particularly when two particles are in proximity, is on the order of 10–20 nm scale for silica microparticles. If this error is similar for our particle system, the corresponding force would be ~0.04–0.08 pN at $\kappa_t = 3.91$ pN/ μm , which is smaller than the error range due to thermal fluctuations of a trapped particle (~30 nm and ~0.12 pN scale, y-direction error bars in Fig. 2d). Additionally, considerable force was not observed in the 10 mM water-only environment (Fig. S3), which demonstrates that such image analysis errors are not significant in our system. Furthermore, for our D–I force measurements and experiments on the dimer formation probability, we bring two particles into close proximity until they come into contact, but we report the force profile prior to contact, which is then used for fitting to determine the power-law exponent (Fig. 2d). At this separation range, we can assume that the two particles are in the same plane, thus minimizing the error due to out-of-plane motion at the minimum separation distance. Therefore, we can ignore any potential image analysis errors due to optical distortion or out-of-plane motion in our system. Note that there are some cases in Fig. 2d where the $\frac{r}{d}$ values are less than 1, which may be attributed to the particle size distribution ($d = 2.96 \pm 0.05$ μm).

Before: **d**, Measured D–I interaction force profile (F_{D-I}) and comparison with the DLVO forces (F_{el} and F_{vdW}). The water phase contains 10 mM NaCl. The red solid line is a fitted curve for F_{D-I} using $F_{fit} = F_0 \left(\frac{d}{r}\right)^b$. The inset indicates the two fitting parameter values for seven pairs.

After (Fig. 2 caption): **d**, Measured D–I interaction force profile (F_{D-I}) and comparison with the DLVO forces (F_{el} and F_{vdW}). The water phase contains 10 mM NaCl. The x-axis of the graph is on a logarithmic scale. The red solid line represents a fitted curve that uses the mean values of the two fitting parameters, $\langle F_0 \rangle = -0.44$ and $\langle b \rangle = 7.18$. The black dotted lines indicate the guideline for $F \sim r^{-7}$. The insets show histograms of the values of the two fitting parameters for 72 pairs. The force profiles display before the paired particles come into contact.

Added statement (p5, SI): Notably, for the pair interaction force measurements reported in Fig. 2d, cases where two particles did not contact each other were only considered. As a result, the possibility of any overlapping effects of the two particles causing artifacts during the image analysis process could be minimized.

3. I presume that interactions were measured by estimating particles' displacements in the potential well of calibrated optical tweezers. If so, this should be explained in the text. How were these calibrations performed? What calibration constants were measured, and what are their uncertainties?

[Type here]

We performed the drag calibration to obtain the trap stiffness and included a detailed description of the method in SI.

Added paragraph (p3, SI):

Drag calibration.

The trap stiffness κ_t was measured by using the drag calibration method, which involved subjecting a trapped particle to drag force by moving a motorized microscope stage (SCAN^{plus} 130×85, Märzhäuser Weltzlar GmbH & Co. KG, Germany) at varying constant velocities between $u = 5\text{--}70\ \mu\text{m/s}$.³⁻⁵ The displacement of the particle Δx from its equilibrium position was caused by the Stokes drag force $F_{Stokes} = 3\pi d\eta_w u$, where η_w is the water viscosity. A plot of F_{Stokes} versus Δx was used to obtain the trap stiffness κ_t via linear regression. This drag calibration was performed at several AOD setting values corresponding to particle positions along the x -direction, where the pair interaction measurements were conducted. The measured κ_t values did not vary significantly with the AOD setting values, as shown in the inset of Fig. S1, and therefore, their mean value of $\langle\kappa_t\rangle = 3.91 \pm 0.07\ \text{pN}/\mu\text{m}$ was used in this study.

To validate the measured κ_t values, we calculated the optical trapping force numerically using the ray optics approximation.⁶ We refer the readers to our previous work for the detailed calculation method.^{5,7} Under the experimental conditions with laser power $P = 7\ \text{mW}$, water and PS refractive indices of $n_1 = 1.326$ and $n_2 = 1.57$, $\text{NA} = 1.2$, and the particle diameter $d = 2.96\ \mu\text{m}$, the calculated trapping force or gradient force F_{trap} was plotted in Fig. S1. The linear regression of the linear regime (light orange region) resulted in $\kappa_t = 3.85 \pm 0.06\ \text{pN}/\mu\text{m}$, which showed excellent agreement with the experimental value.

Added Fig. S1 in SI:

[Type here]

Fig. S1. Optical trap calibration. Numerical calculations of the optical trapping force F_{trap} as a function of lateral displacement Δx at a laser power of $P = 7$ mW. The linear fit in the light orange region estimates the trap stiffness about $\kappa_t = 3.85 \pm 0.06$ pN/ μ m. The inset shows the results of drag calibration experiments conducted at various set values of AOD, which correspond to the x -positions of the optical trap used in the pair interaction measurements. Each data point in the inset plot indicates the mean value of three-independent runs of the drag calibration.

4. If forces were measured by monitoring a particle's displacement in its trap, how large are the trapped particles' thermal fluctuations? In light of these fluctuations, how many particle-separation measurements contribute to each point in the reported force-separation curves?

When we measure the pair interaction force, we bring the P_2 particle stepwise toward the P_1 particle to minimize any hydrodynamic effects. During each step, the particle remains in place for ~ 30 s, and a total of ~ 900 images are averaged to obtain their positions. The typical fluctuation of a particle trapped with $\kappa_t = 3.91$ pN/ μ m is ~ 30 nm, which corresponds to ~ 0.12 pN. We measured the force for 72 particle pairs, as shown in the revised Fig. 2d, where the error bars of data points along the y-axis direction represent this fluctuation.

Revised Fig. 2d:

Fig. 2. Direct measurements of the D-I interaction force. d, Measured D-I interaction force profile (F_{D-I}) and comparison with the DLVO forces (F_{el} and F_{vdW}). The water phase contains 10 mM NaCl. The x-axis of the graph is on a logarithmic scale. The red solid line represents a fitted curve that uses the mean values of the two fitting parameters, $\langle F_0 \rangle = -0.44$ and $\langle b \rangle = 7.18$. The black dotted lines indicate the guideline for $F \sim r^{-7}$. The insets show

[Type here]

histograms of the values of the two fitting parameters for 72 pairs. The force profiles display before the paired particles come into contact.

5. How do we know that the optical tweezer does not affect measured interparticle interactions?
5.a) Trapped particles can be attracted to their neighbors' traps, particularly at small separations. How do we know that this can be ignored? If it cannot be ignored, how was corrected?

5.b) Light scattered by the trapped particle can give rise to inter-particle forces. The need to measure and correct for such effects has been discussed, for example, in Crocker JC, Matteo JA, Dinsmore AD, Yodh AG. Entropic attraction and repulsion in binary colloids probed with a line optical tweezer. *Physical Review Letters* 82, 4352 (1999).

Other groups have avoided this problem by turning off the optical tweezers during an interaction measurement, with relevant references including

Crocker JC, Grier DG. Microscopic measurement of the pair interaction potential of charge-stabilized colloid. *Physical Review Letters* 73, 352 (1994); and

Sainis SK, Germain V, Dufresne ER. Statistics of particle trajectories at short time intervals reveal fN-scale colloidal forces. *Physical Review Letters* 99, 018303 (2007).

5.c) Light absorbed from the optical tweezer can heat the trapped particle and the surrounding medium. Heating can change charged particles' charges and can induce flows in the surrounding fluid. How do we know that optically-induced heating can be neglected? This also is why it is necessary to specify the wavelength and power of the trapping laser.

Above comments refer to potential artifacts that may arise from the use of optical tweezers at close distances and image analysis methods. In our experiment, we used an adequately weak time-sharing optical trap (with a laser power of ~ 7 mW per trapped particle), which suggests that any artifacts related to light scattering effects or heat generation from the optical tweezer would be minimal. As previously mentioned, we did not consider force data when two particles were in contact and out of plane. Furthermore, the fact that the pair force before contact is negligible in the 10 mM NaCl water-only environment (Fig. S3) serves as evidence that any artifacts from the optical tweezer can be disregarded in our experimental setup. If they do exist, they would be smaller than the thermal fluctuation level of ~ 0.12 pN, which would not significantly affect the results of the power-law fitting of the D-I interaction force.

Revised Fig. S3:

[Type here]

Fig. S3. Pair interaction forces for nine different pairs suspended in 10 mM NaCl water.

The grey circles represent the force profile for each pair, with the z-axis error bars indicating thermal fluctuations observed while holding the particles with optical tweezers. The red circles represent the average force profile over the nine pairs, with the error bars indicating the corresponding standard deviation. The x-axis is shown in log scale. The inset provides a magnified view of the force profile near separations.

5.d) The proximity of the oil-water interface can influence the optical force field through the light that it scatters. How do we know that the trap's position and stiffness were not affected by the proximity of the interface?

We thank the reviewer for bringing up this critical issue. In our experiment, the position and stiffness of the optical trap were negligibly affected by the proximity of the oil-water interface. The pair interaction force is calculated by measuring the displacement Δx of P_2 particle and using the equation $F = \kappa_t \Delta x$. Therefore, the optical trap in which P_1 particle is held is only used for the purpose of holding the particle at the interface. The three-phase contact angle between the sessile drop and the bottom coverslip is $\sim 69^\circ$. The angle between the focal plane of the trapped particle and the fluid interface is not expected to change significantly because the particles are located about 40–50 μm from the coverslip, which is sufficiently low. The objective used in the experiment has $\text{NA} = 1.2$, and the maximum half angle of the cone-like laser beam is $\sim 64.5^\circ$ [Park and Furst, Langmuir, 24, 13383 (2008)], which is smaller than the contact angle, $\sim 69^\circ$. Thus, even if the trapped particle is in proximity to the interface, the scattering of the Gaussian beam by the fluid interface can be neglected in our experimental condition. We clarified this point in the revised manuscript.

Added paragraph (p20): Note that the oil-water interface of the sessile drop formed a contact angle of $\sim 69^\circ$ with the bottom coverslip. The pair interaction measurement was conducted at a distance of approximately 40–50 μm from the coverslip surface. Given that the height of the sessile drop apex from the coverslip was at a millimeter scale, the angle between the focal plane

[Type here]

at which P_1^i was located and the interface could be assumed to be the same as $\sim 69^\circ$. Although this geometry might affect the D–I interaction due to the non-parallel approach of P_2 to the dipole formed on P_1^i , it could help prevent unquantifiable artifacts resulting from the use of laser tweezers. The highly focused laser beam from the objective lens with $NA = 1.2$ might scatter at the oil–water interface when it trapped a particle near the interface, leading to potential errors in force measurements. However, in the current setup, the objective lens generated a maximum half angle of $\sim 64.5^\circ$ for the con-like laser beam,⁴³ which was smaller than the contact angle of 69° between the interface and coverslip, thereby eliminating such potential artifacts.

6. The data in Fig. 2(d) are supposed to justify the paper’s principal conclusions regarding the nature of the interparticle forces. The current presentation, however, is not convincing. Most data points are obtained at such large separations that interparticle forces are negligible. Just a few measurements at very small separations are used to infer the functional form of the interaction force. The manuscript should include a version of this plot that zooms in on the region near contact so that the reader can assess the validity of the comparison with the Debye force law. A log-log plot would be helpful to assess how well the data are described by power-law scaling and the exponent of that scaling law.

We measured the D–I interaction force for 72 particle pairs, and as suggested by the reviewer, we presented the force profiles on a log scale for the x -axis in the revised Fig. 2d.

Revised Fig. 2d:

Fig. 2. Direct measurements of the D–I interaction force. d, Measured D–I interaction force profile (F_{D-I}) and comparison with the DLVO forces (F_{el} and F_{vdW}). The water phase contains 10 mM NaCl. The x -axis of the graph is on a logarithmic scale. The red solid line represents a fitted curve that uses the mean values of the two fitting parameters, $\langle F_0 \rangle = -0.44$ and $\langle b \rangle = 7.18$. The black dotted lines indicate the guideline for $F \sim r^{-7}$. The insets show

[Type here]

histograms of the values of the two fitting parameters for 72 pairs. The force profiles display before the paired particles come into contact.

7. The most relevant range of separations for measuring the interparticle force also is the range of separations where the optical tweezer itself can mediate interactions between the particles and where the presence of the interfacial particle can modify the potential energy well of the optical tweezer. It also is the range where diffraction introduces overlap errors in image-based particle tracking. It also is the range where out-of-plane motions can be most significant and also most difficult to measure. If, as I suspect, just a few data points support the interpretation of the experiment, then particular care will be needed to exclude all such artifacts.

We believe that this comment has already been addressed in our previous responses.

REVIEWER COMMENTS

Reviewer #1 (Remarks to the Author):

Taking into account the comments from the other reviewers and I, I believe that the revised manuscript has addressed most of the points raised in the first round.

My major concern is about the revised Fig.2d. On the one hand, the authors included more information than in the original manuscript, which improved our ability to assess the significance of the results. On the other hand, it is impossible to qualitatively assess the appropriateness of the $\sim r^{-7}$ dependence of the force. In this regard, I would have liked to see (perhaps as an additional SI figure) a plot in which BOTH scales are logarithmic (the force might be written in absolute value). From the spread of exponents observed in the inset histogram, I argue that this plot would look messy though. Perhaps the authors could plot the curve obtained by averaging over all the particles?

Some minor suggestions:

- 1) It might be worth attempting to compute the induced dipole from the simulations, for which all the needed data are available. The tricky part is to define the region of space where to compute it, but this computation might further enforce the point being made (existence of induced dipole in the case with interface as compared to the control)
- 2) What is the Young modulus obtained from the fitting with the Euler-Bernoulli formula in Fig.6? How does it compare with the computed value?
- 3) There are no error bars in Fig.5e
- 4) page 18, line 331: I guess the authors refer to Fig. S13 here?

Reviewer #2 (Remarks to the Author):

The authors have convincingly addressed all my comments. I can now recommend the publication of their study in Nature Communications.

Reviewer #3 (Remarks to the Author):

The revised manuscript provides enough information about the experimental implementation for a knowledgeable reader to reproduce the work. The principal experimental observations support the interpretation that a colloidal particle embedded in an oil-water interface exerts an attractive force on nearby particles in the water phase, and that the attraction can be transmitted to create a chain of bound particles. The experiments on interfacial colloids are complemented by control measurements on particles entirely immersed in water. No attractions are evident in those control experiments. The discovery of interface-induced colloidal attractions and collective behavior induced by those attractions is compelling and merits publication in Nature Communications.

The manuscript also reports measurements of the interparticle force based on imaging measurements of particles in time-shared optical tweezers. The revised manuscript compares the interaction force between interfacial and water-borne particles in Fig. 2d with control measurements in Fig. S3. Long-ranged attractions in the control measurement appear to be at least a factor of ten weaker than in the main data set. This substantially addresses my initial concern that the attractions reported in Fig. 2d could be ascribed to some combination of imaging artifacts, out-of-plane motion and forces exerted by the optical traps.

I still have misgivings about the quantitative results obtained from the imaging studies, but not to the extent that I would want to delay publication. I similarly wonder if alternative

mechanisms could be developed for the observed attraction. Wicking of oil between the interfacial spheres, for example, could mediate a strong attractive interaction. This mechanism, moreover, could account for the time required to create a bound interfacial pair and to grow a chain. This also might explain why the pair-formation process is not 100% reproducible. Having given it some thought, I have concluded that such speculations would be better left for future studies.

REVIEWER COMMENTS

Reviewer #1 (Remarks to the Author):

Taking into account the comments from the other reviewers and I, I believe that the revised manuscript has addressed most of the points raised in the first round.

We greatly appreciate the reviewer's helpful comments, which have significantly improved our paper in the first round. In response to the additional comments, we provide the following answers:

My major concern is about the revised Fig.2d. On the one hand, the authors included more information than in the original manuscript, which improved our ability to assess the significance of the results. On the other hand, it is impossible to qualitatively assess the appropriateness of the $\sim r^{-7}$ dependence of the force. In this regard, I would have liked to see (perhaps as an additional SI figure) a plot in which BOTH scales are logarithmic (the force might be written in absolute value). From the spread of exponents observed in the inset histogram, I argue that this plot would look messy though. Perhaps the authors could plot the curve obtained by averaging over all the particles?

As suggested by the reviewer, we have added a log-log plot as an inset in the revised Fig. 2d, after taking the absolute value of the measured force. Since the force values approach zero at long-range separation, we plotted the force only in the short-range separation to clearly demonstrate the r^{-7} dependence, and added a solid red line representing $\langle F_{fit} \rangle \sim r^{-7.18}$. As shown in the modified plot below, it is evident that the experimental force profiles align well with r^{-7} . While averaging the force data for all particles would not be difficult, it would not clearly exhibit the r^{-7} dependence.

Before (Fig. 2d):

[Type here]

Fig. 2. Direct measurements of the D–I interaction force. **d**, Measured D–I interaction force profile (F_{D-I}) and comparison with the DLVO forces (F_{el} and F_{vdW}). The water phase contains 10 mM NaCl. The x-axis of the graph is on a logarithmic scale. The red solid line represents a fitted curve that uses the mean values of the two fitting parameters, $\langle F_0 \rangle = -0.44$ and $\langle b \rangle = 7.18$. The black dotted lines indicate the guideline for $F \sim r^{-7}$. The insets show histograms of the values of the two fitting parameters for 72 pairs. The force profiles display before the paired particles come into contact.

After (Fig. 2d):

[Type here]

Fig. 2. Direct measurements of the D–I interaction force. **d**, Measured D–I interaction force profile (F_{D-I}) and comparison with the DLVO forces (F_{el} and F_{vdW}). The water phase contains 10 mM NaCl. The x-axis of the graph is on a logarithmic scale. The red solid line represents a fitted curve that uses the mean values of the two fitting parameters, $\langle F_0 \rangle = -0.44$ and $\langle b \rangle = 7.18$. The black dotted lines indicate the guideline for $F \sim r^{-7}$. The force profiles display before the paired particles come into contact. The left two inset plots display histograms of the values of the two fitting parameters for 72 pairs. The inset on the right represents a log-log plot of $|F|$ versus r/d for short-range separation.

Some minor suggestions:

1) It might be worth attempting to compute the induced dipole from the simulations, for which all the needed data are available. The tricky part is to define the region of space where to compute it, but this computation might further enforce the point being made (existence of induced dipole in the case with interface as compared to the control)

To answer the reviewer's comment, we have computed the variation in the dipole strength of P_2 as it moves toward the P_1^i particle attached to the oil-water interface, corresponding to the case of Fig. 4a and 4b. As shown in Fig. R1, the highest dipole strength was calculated when P_2 was in the D-I dominant region. On the other hand, in the intermediate and vdW dominant regions, the dipole strength was similar to that prior to the 200 ns region (Fig. 4b). This allows us to indirectly verify that the induced dipole moment was formed in the interfacial environment. Note that since the charged components in our system are not stationary and the partition criteria of the region are difficult to define, we calculated the dipole strength considering all charged components in the system. This resulted in small, but same degrees of error in the estimated dipole strengths.

Fig. R1. Variation in the dipole strength of P_2 as it approaches toward the P_1^i particle (see Fig. 4a and 4b).

2) What is the Young modulus obtained from the fitting with the Euler-Bernoulli formula in Fig.6? How does it compare with the computed value?

We appreciate the reviewer's suggestion and have found that the Young's modulus value, derived directly from fitting with the Euler-Bernoulli equation, aligns very well with the value

[Type here]

previously obtained through the single-bond rigidity. We have added this result to the revised SI.

Added paragraph (p9, SI): Alternatively, the Young's modulus could be directly determined by fitting the colloidal chain profile with the Euler-Bernoulli beam equation. For instance, in the case of maximum bending (IV in Fig. 6e), its shape was fitted using the equation $y = \frac{F}{EI} \left(L \frac{(x-x_0)^2}{2} - \frac{(x-x_0)^3}{6} \right) + y_0$, where $L = s$, which resulted in $\frac{F}{EI} = 0.0025 \pm 0.0003 \mu\text{m}^{-2}$. By substituting the value of F with $F_{bend} = 1.78 \text{ pN}$ at $\delta = 4.25 \mu\text{m}$ and using the relationship of $I = \frac{\pi a_c^4}{4}$, the Young's modulus was $E_{chain,PS} = 0.03 \text{ GPa}$. This result aligns well with the previously obtained value of 0.05 GPa from the single-bond rigidity.

3) There are no error bars in Fig.5e

We thank the reviewer for the valuable comment and apologize for the unclear presentation of number density in Fig. 5e. To provide a clearer presentation, we ran three independent simulations to acquire the average values along with the corresponding error bars.

Before (Fig. 5e):

Fig. 5. CGMD simulations for the trimer formation. e, Number density difference of Cl^- ions in the left and right regions between the two particles.

After (Fig. 5e):

[Type here]

Fig. 5. CGMD simulations for the trimer formation. e, Number density difference of Cl^- ions in the left and right regions between the two particles. The error bar indicates three independent simulation runs, and the inset represents the analyzed region.

4) page 18, line 331: I guess the authors refer to Fig. S13 here?

We thank the reviewer for pointing out this error, which was corrected in the revised manuscript.

Reviewer #2 (Remarks to the Author):

The authors have convincingly addressed all my comments. I can now recommend the publication of their study in Nature Communications.

Reviewer #3 (Remarks to the Author):

The revised manuscript provides enough information about the experimental implementation for a knowledgeable reader to reproduce the work. The principal experimental observations support the interpretation that a colloidal particle embedded in an oil-water interface exerts an attractive force on nearby particles in the water phase, and that the attraction can be transmitted to create a chain of bound particles. The experiments on interfacial colloids are complemented by control measurements on particles entirely immersed in water. No attractions are evident in those control experiments. The discovery of interface-induced colloidal attractions and collective behavior induced by those attractions is compelling and merits publication in Nature Communications.

The manuscript also reports measurements of the interparticle force based on imaging measurements of particles in time-shared optical tweezers. The revised manuscript compares the interaction force between interfacial and water-borne particles in Fig. 2d with control measurements in Fig. S3. Long-ranged attractions in the control measurement appear to be at least a factor of ten weaker than in the main data set. This substantially addresses my initial concern that the attractions reported in Fig. 2d could be ascribed to some combination of

[Type here]

imaging artifacts, out-of-plane motion and forces exerted by the optical traps.

We thank the reviewer for acknowledging our responses, particularly regarding the potential for minor experimental errors arising from optical trapping under our experimental conditions.

I still have misgivings about the quantitative results obtained from the imaging studies, but not to the extent that I would want to delay publication. I similarly wonder if alternative mechanisms could be developed for the observed attraction. Wicking of oil between the interfacial spheres, for example, could mediate a strong attractive interaction. This mechanism, moreover, could account for the time required to create a bound interfacial pair and to grow a chain. This also might explain why the pair-formation process is not 100% reproducible. Having given it some thought, I have concluded that such speculations would be better left for future studies.

The reviewer's suggestion regarding the "Wicking of oil between the interfacial spheres" is intriguing. As we interpret it, this implies that when a P_2 particle in water approaches a P_1^i particle attached to the interface to form a $P_1^i - P_2$ dimer, decane on the opposite side might permeate or reach between the two particles, potentially influencing the dimer formation. As the reviewer suggested, it would indeed be worthwhile to evaluate the existence or absence of this effect in an independent follow-up study. We greatly appreciate this interesting suggestion for subsequent research.

REVIEWERS' COMMENTS

Reviewer #1 (Remarks to the Author):

The Authors have fully addressed the comments of the three Reviewers. I am glad to recommend the publication of this nice work in its current status on Nature Communications.

Reviewer #3 (Remarks to the Author):

I recommend that the revised manuscript be published in Nature Communications.